A comprehensive review of ball detection techniques in sports

Moreira Cristiano 1
Ferreira Lino 1 2
http://orcid.org/0000-0002-4383-0472 Coelho Paulo Jorge 1 3 paulo.coelho@ipleiria.pt
1 Electrical Engineering Department, Polytechnic University of Leiria , Leiria , Portugal
2 Electrical Engineering Department, Instituto de Telecomunicações , Leiria , Portugal
3 Electrical Engineering Department, Institute for Systems Engineering and Computers at Coimbra (INESC Coimbra) , Coimbra , Portugal
Alatas Bilal
Electronic publication date: 2025 Aug 12
Publication date: 2025
Volume: 11
Electronic Location ID: e3079
Received 2025 Mar 7; Accepted 2025 Jul 4
Copyright: © 2025 Moreira et al.
Copyright year: 2025
Copyright holder: Moreira et al.
License: This is an open access article distributed under the terms of the Creative Commons Attribution License, which permits unrestricted use, distribution, reproduction and adaptation in any medium and for any purpose provided that it is properly attributed. For attribution, the original author(s), title, publication source (PeerJ Computer Science) and either DOI or URL of the article must be cited.
License URL: https://creativecommons.org/licenses/by/4.0/

Keywords: Object detection, Computer vision, Deep learning, Ball detection, Sports

Funding: FCT/MEC FEDER-PT2020 UIDB/00308/2020 This work was funded by FCT/MEC through national funds and co-funded by the FEDER-PT2020 partnership agreement under the project UIDB/00308/2020 (DOI 10.54499/UIDB/00308/2020). The funders had no role in study design, data collection and analysis, decision to publish, or preparation of the manuscript.

==============================
Detecting balls in sports plays a pivotal role in enhancing game analysis, providing real-time data for spectators, and improving decision-making and strategic thinking for referees and coaches. This is a highly debated and researched topic, but most works focus on one sport. Effective generalization of a single method or algorithm to different sports is much harder to achieve. This article reviews methodologies and advancements in object detection tailored to ball detection across various sports. Traditional computer vision techniques and modern deep learning methods are visited, emphasizing their strengths, limitations, and adaptability to diverse game scenarios. The challenges of occlusion, dynamic backgrounds, varying ball sizes, and high-speed movements are identified and discussed. This review aims to consolidate existing knowledge, compare state-of-the-art detection models, highlight pivotal challenges and possible solutions, and propose future research directions. The article underscores the importance of optimizations for accurate and efficient ball detection, setting the foundation for next-generation sports analytics systems.

Introduction

Understanding the ball’s movement in sports analytics is integral to game analysis, strategy formulation, and audience engagement. The ball often serves as the focal point around which key events unfold, making its detection and tracking a critical task. However, the dynamic and unpredictable nature of sports introduces numerous challenges, including complex backgrounds, occlusions by players, small object sizes, and the high-speed movement of the ball. These challenges require advanced detection techniques capable of operating with high accuracy in real-time.

The evolution of object detection methodologies, from traditional computer vision (CV) techniques such as colour segmentation and background extraction, to machine learning and feature-based algorithms like histogram of oriented gradients (HOG) (Dalal & Triggs, 2005) and scale-invariant feature transform (SIFT) (Lowe, 1999), to deep learning-based frameworks such as faster region-based convolutional neural networks (R-CNN) (Ren et al., 2017), you only look once (YOLO) (Redmon et al., 2016), single shot detector (SSD) (Liu et al., 2016), RetinaNet (Lin et al., 2017), and Mask R-CNN (He et al., 2017) has significantly improved detection capabilities in both performance and efficiency. Despite these advancements, the unique constraints of sports environments demand further innovation. Factors such as varying ball sizes and colours across sports, high object speeds, lighting conditions, different fields or courts, and the need for seamless integration with tracking systems underline the need for tailored solutions.

This article reviews the state-of-the-art techniques in ball detection across sports, providing a concentrated and comprehensive overview of existing challenges, tools, and applications. This review aims to bridge gaps in the literature and inspire future research in this domain by categorizing approaches based on their underlying methods and sports-specific adaptations. The subsequent sections explore traditional and deep learning techniques, benchmark datasets, and performance metrics, culminating in insights for developing robust ball detection systems suitable for diverse sporting scenarios and hardware. The present article also strives to give useful insights on the topic of object detection, especially applied to sports, earned from meticulously analysing and reviewing these various sources of information.

The proposed article aims to gather and concentrate information on object detection applied to sports from reliable sources, and give valuable insight on the topic, earned from meticulously analysing and studying said information.

This article is divided into six sections: Related Work, details articles that also review object detection applied to sports balls. Methodology presents research questions that this review pretends to answer and explains the strategy behind the research, while Results, introduces and explains the articles that resulted from the research. Discussion presents a summary of what was gathered from the articles researched and brings about a discussion on the most valuable information. Finally, the Conclusions brings this review to a close with conclusions on what was learned and contributed.

Related work

This section analyses some of the most recent and relevant reviews on this topic. Object detection has been, and continues to be, a highly debated and researched topic. The sheer number of research articles and conference articles that appear when searching for the term “object detection” is evident. The number of reviews focusing solely on spheric objects is significantly more limited. Most reviews that focus on sports ball detection also concentrate on a single sport or one or two specific detection methods, leaving little room for newer or less popular techniques to shine. This article compiles recently published reviews (2015–2024) on ball detection, regardless of method or application, as long as their contents may be useful to ball detection in the sports industry.

The number of articles found reviewing or surveying ball detection applied specifically to sports was quite low. Most of this low amount is specifically aimed at football applications. Because of this, this review includes a couple of additional articles that either analyse ball detection but do not aim to apply it to sports or analyse player detection in some type of sport.

Mishra, Chandra & Jaiswal (2015) reviews works on player tracking in generic sports videos, weighing challenges such as occlusion and fast movements, and highlighting the evolution of tracking methods from 2003 to 2014. This review focuses solely on traditional techniques, including background subtraction, particle filters, and feature-based methods, applied across diverse sports datasets.

Using handball-oriented datasets, the article by Burić, Pobar & Ivasic-Kos (2018a) compares Mask R-CNN and YOLOv2 on ball detection. Different training configurations and datasets were employed, and YOLO emerged as the preferred method for real-time use.

In another article by the same author, Burić, Pobar & Ivašić-Kos (2018b) evaluates object detection methods for sports videos, once again focusing on action recognition in handball scenes. It compares Mask R-CNN, YOLO, and a mixture of Gaussian (MOG) and attempts to address challenges such as fast movements and diverse backgrounds.

Badami et al. (2018) review analyses of football videos, detecting various players’ body parts and the ball to identify poses, actions, and events. This work proposes an automated refereeing system that utilizes computer vision techniques, such as HOG, to address the limitations of human referees and reduce controversies in football.

Taken together, these four studies mark the field’s transition from rule-based computer-vision pipelines (Mishra, Chandra & Jaiswal, 2015) toward the first CNN-powered, real-time detectors (Burić, Pobar & Ivasic-Kos, 2018a), highlighting a simultaneous gain in speed (YOLOv2) and in segmentation granularity (Mask R-CNN).

The survey, authored by Kamble, Keskar & Bhurchandi (2019), analyses techniques for detecting and tracking balls in generic sports. It differentiates and categorizes methods for ball detection and tracking separately. For ball detection, the methods are categorized into four main areas: colour, size, position, and motion. Kalman filter, particle filters, and trajectory are used to categorize and compare tracking efficiency. Although this work only surveys traditional computer vision techniques, the authors suggest that “the future lies in the right employment of deep learning techniques (…)”. Challenges like occlusion, rapid motion, and lighting variations are highlighted. Evaluations utilize metrics such as precision and recall, with datasets spanning various sports and camera setups.

Deepa et al. (2019) investigates real-time tennis ball tracking using YOLO, SSD, and Faster R-CNN, aiming to create a cost-effective alternative to systems like Hawk-Eye for analysing ball trajectories. The models were evaluated based on their accuracy and computational power required to detect and calculate metrics such as toss height, toss distance, serve speed, and hit speed, using diverse datasets collected from multiple cameras.

The article by Cuevas, Quilón & García (2020) surveys techniques and applications for football video analysis, with a focus on player and ball tracking for event detection. It categorizes various methods based on audiovisual information, such as support vector machines (SVM), neural networks (NN), hidden Markov models (HMM), and probabilistic-based approaches. Methods based on external sources, such as webcasting and social networks, are also mentioned. The authors give great importance to the real-time applicability and various challenges encountered in ball detection, such as its small size, occlusion due to players, misdetections when the ball stands on top of a marked line on the field, and changes in the ball’s size, shape, colour, and speed.

Jagadeesh, R & S (2023) surveyed deep learning techniques, such as CNN and transfer learning, to classify cricket shots, not in real-time. The research evaluates models using metrics such as Precision, Recall, and confusion matrices, with a dataset of cricket shot images that have been augmented and pre-processed for consistency. Results highlight the effectiveness of transfer learning models, particularly visual geometry groups (VGG), in accurately detecting shot types. Using a pre-trained 19-layer VGG-19 model, the authors claim that this deep architecture is capable of recognizing various objects and patterns.

The review by Şah & Direkoğlu (2023) evaluates and compares player detection methods and techniques in field sports. The authors discuss conventional computer vision techniques, including HOG, SVM, and heat diffusion (HD), as well as CNN architectures with various image representations, such as gray, red-green-blue (RGB), shape information image (SIM), and polar-transformed shape information image (PSIM). Transfer learning was also compared using deep networks, such as Faster R-CNN, SSD, and YOLO. The study utilizes two field hockey datasets to evaluate precision, recall, and F-measure under consistent settings, taking into account occlusion and overlap ratios. Their results state that conventional methods and CNNs with SIM and PSIM image representations are preferred over deep networks when treating low-resolution images, which is common in field sports videos.

Akan & Varlı (2023) explores deep learning in football video analysis, addressing both player and ball tracking for real-time event detection. They review classical computer vision techniques and deep learning-based methods, highlighting the superiority of the latter in capturing high-level semantic information compared to traditional blob-based and feature-based approaches. The authors conclude that deep learning models, such as CNN and recurrent neural networks (RNN), overcome the drawbacks of classical approaches, performing with better efficiency and adaptability to various scenarios and complexities.

Yang et al. (2024a) review advancements in football player detection and tracking, emphasizing the growing demand for automated football video analysis. The study examines traditional methods and modern approaches using multi-object tracking (MOT). It evaluates different models using performance metrics such as detection accuracy, multi-object tracking accuracy (MOTA), and high-order tracking accuracy (HOTA).

Markappa, O’Leary & Lynch (2024) evaluates various YOLO models and versions in detecting football-related entities like players, referees, and balls. Using the SoccerNet dataset (Giancola et al., 2018), the training and assessment of the models in various Precision metrics and across multiple epochs revealed superior performance by YOLOv8 and YOLOv9.

To better articulate the novelty and analytical depth of this review, the authors introduce a formal taxonomy that organizes prior work across three core dimensions: object of detection, application scope, and technical approach. These categories are reflected in Table 1 and are further clarified below:

Object of detection: We distinguish between articles that study sports ball detection (DB: detects balls) and those that focus exclusively on ball detection (FB: focuses on balls). This distinction highlights the specificity of the work and helps identify gaps where ball-only detection is understudied;

Application scope: This dimension captures the review’s generalization and domain focus. Articles are marked based on whether they apply to sports contexts (AS: applied to sports) and whether they span multiple sports rather than focusing on a single discipline (GS: generalized sports). This allows us to assess the transferability of techniques across different sports;

Technical approach: We classify the detection methodologies as traditional computer vision and machine learning techniques (CV: computer vision and machine learning) or deep learning-based methods (DL: deep learning). This distinction reflects the field’s evolution and helps identify which methods are underrepresented in reviews.

Table 1 Listing and categorizing existing articles in the ball detection applied to sports.

Article	DB	FB	AS	GS	CV	DL	
Survey on various techniques of tracking (Mishra, Chandra & Jaiswal, 2015)		✓	✓	✓			
Ball detection using YOLO and Mask R-CNN (Burić, Pobar & Ivasic-Kos, 2018a)	✓	✓	✓			✓	
Object detection in sports videos (Burić, Pobar & Ivašić-Kos, 2018b)	✓		✓		✓	✓	
Review on video refereeing using computer vision in football (Badami et al., 2018)	✓		✓		✓		
Ball tracking in sports: a survey (Kamble, Keskar & Bhurchandi, 2019)	✓	✓	✓	✓	✓		
Comparison of YOLO, SSD, Faster RCNN for real-time tennis ball tracking for action decision networks (Deepa et al., 2019)	✓	✓	✓			✓	
Techniques and applications for football video analysis: a survey (Cuevas, Quilón & García, 2020)	✓		✓		✓		
Cricket shot detection using deep learning: a comprehensive survey (Jagadeesh, R & S, 2023)	✓		✓			✓	
Review and evaluation of player detection methods in field sports (Şah & Direkoğlu, 2023)			✓	✓	✓	✓	
Use of deep learning in football videos analysis: survey (Akan & Varlı, 2023)	✓		✓			✓	
A survey on football player detection and tracking with videos (Yang et al., 2024a)			✓		✓	✓	
A review of YOLO models for football-based object detection (Markappa, O’Leary & Lynch, 2024)	✓		✓			✓	
This review	✓	✓	✓	✓	✓	✓	

This taxonomy allows for a structured comparison between existing reviews. It underscores where this article extends beyond the scope of previous works, particularly by being one of the few to offer a ball-specific, sport-agnostic, and DL-inclusive review.

Most review literature does not generalize its work findings to different sports. Doing so will increase its usefulness to a broader audience. While someone looking into football ball detection techniques might be better off researching reviews that specifically target football, since there are many, most other sports applications would benefit more from a generalized view of what is available for ball detection in sports. The proposed review intends to do just that: review all sorts of algorithms to search for one that can perform well in any ball-playing sport. Also, these do not specialize solely in ball detection. It leaves room for improvement in systems designed solely for ball identification and tracking, as training them for a specific task will significantly enhance their effectiveness. Some methods or techniques may be more suitable than others for specific objects, depending on factors such as object speed, size, and complexity. This means that a specific detector that excels in both player and ball detection in a football game, for example, might not be the best detector for football ball detection only. That again differentiates this review, as it strives to achieve generalized ball-only detection.

The proposed review aims to gather and explore information on object detection applied to sports from reliable sources, providing valuable insights into the topic through meticulous analysis and study of the information. The article focuses on detecting sports balls but does not specialize in a single specific sport, aiming to give generalized insight into sports as a whole and the challenges one model might face, regardless of the sport to which it is applied. This review also aims to address the limitations identified in other articles. The survey by Kamble, Keskar & Bhurchandi (2019) does most of the mentioned goals, but unlike this review, it does not research deep learning methods. These techniques have been evolving rapidly and are something that cannot be overlooked at the current time. The review by Şah & Direkoğlu (2023) seems to do what this review intends to, but it is applied to player detection instead of ball detection.

What further distinguishes this review is its broad scope, encompassing ball detection across 12 different sports, including less commonly studied ones such as boccia, padel, and rugby. Unlike prior surveys focusing on a single sport (often football), this work compiles methods and insights from various disciplines, making it more broadly applicable. This review addresses the fragmentation by intentionally selecting articles that extend beyond mainstream sports and balancing coverage across traditional computer vision and modern deep learning approaches. It delivers a more inclusive and sport-agnostic perspective.

The following section will outline the methodology of this review, examining its objectives and the approach taken during the investigation.

Methodology

This chapter will outline the objectives of the proposed review and the methodologies employed in the research. The initial and main research was conducted on October 17, 2024, and an additional search was performed on May 22, 2025, aiming to update and identify additional relevant and recent studies. The search was conducted using predetermined inclusion and exclusion criteria and well-defined terms to gather articles that seemed relevant from various databases. The following subsections will outline the research questions this comprehensive review aims to address and the search strategy employed to gather the reviewed manuscripts.

Research questions

The following are the primary inquiries this review seeks to answer: RQ1: What are the most popular methods in round object detection?

RQ2: What are the advantages and disadvantages of these methods?

RQ3: What are the major challenges tied to object detection in sports?

RQ4: What methods or combination of methods and techniques can be considered the best for generic ball detection applied to sports?

This review aims to thoroughly examine these questions and provide insightful answers to each, offering multiple options for various scenarios.

Search strategy

The initial search utilizes well-defined search terms to gather relevant articles from various databases. The removal of duplicates and unrelated or incomplete articles follows this initial step. A manual screening of the remaining articles removes any unwanted articles. From the results, a qualitative analysis is performed to identify relevant articles, culminating in a small group of high-quality articles that assess the most recent developments in object detection. Various searches were made in Google Scholar through generic but relevant keywords and phrases, such as “Sports balls object detection”, for example. Articles found within Google Scholar from these searches are indexed in other databases. The databases from which the articles were gathered for this review were obtained from IEEE Xplore, Springer, Scopus, and ArXiv, which are the most representative databases for computer vision and computer science. These researches followed the following criteria:

Inclusion criteria 1. Articles written in English;

2. Peer-reviewed manuscripts;

3. Full text access to manuscript;

4. Articles published from 2015 onward;

5. Articles that follow the keywords (Object AND Detection AND Sports) AND (Balls OR Circular OR Circle).

Exclusion criteria 1. Articles that are categorized as reviews or surveys;

2. Articles with an invalid DOI (Digital Object Identifier);

3. Articles that do not present experimental results.

A total of 927 studies came up from the initial search: 182 from IEEE Xplore, 458 from Springer, 236 from Scopus, and 51 from other sources from generic searches in Google Scholar. From this relatively large group of articles, a pre-screening was done, immediately excluding a significant amount of articles for a multitude of reasons: 39 were rapidly removed for duplication between databases, 139 articles were excluded for not being in the computer vision or engineering’s subject areas (disciplines in Springer), 186 for the publication date (articles published before 2015), four for their language (articles not written in English), and 43 articles were removed for being reviews or surveys.

After this initial filtering, 516 studies underwent a manual screening, eliminating 479 articles. This procedure, undergone by all authors of this review, aimed to assess the quality and reliability of a article. Starting with an inspection of the article’s title and abstract, the assessment considered the relevance of the study and its objectives to the topic at hand, adherence to writing standards, structural integrity, the display of results, and whether evaluations and assessments were conducted. After this screening process, 37 articles remained, of which six could not be retrieved due to the lack of access, incorporating 31 relevant studies in the review.

An updated research was performed on May 22, 2025, aiming to gather more recent studies. This research focused on the indexing engine Google Scholar and the ArXiv database. The search in ArXiv included the terms “sports”, “object” and “detection” simultaneously, which resulted in a total of 81 articles. A pre-screening was performed, limiting the search to the subject of computer science, limiting the publication dates to 2020 onwards, and excluding reviews and surveys. From this pre-screening, 62 articles remained, 61 of which were from ArXiv, and one relevant article was found in IEEE Xplore through Google Scholar’s indexing. The articles underwent a manual screening, of which 55 were removed. The remaining seven articles, published between 2021 and 2024, were added to the review, bringing the total to 38 articles incorporated into the meta-analysis of this review. A flowchart illustrating this entire process is depicted in Fig. 1.

Figure 1 Flowchart of the research done.

Summing up, all the articles gathered for reviewing consist of one (2.63%) article from 2015, four (10.53%) articles from 2018, seven (18.42%) articles from 2019, three (7.89%) articles from 2020, six (15.79%) articles from 2021, four (10.53%) articles from 2023 and 13 (34.21%) articles from 2024, amounting to a total of 38 articles reviewed.

Unlike prior reviews that often rely on a narrow or informal selection of studies, our approach involved a well-defined set of inclusion and exclusion criteria applied across multiple major databases. This process enabled us to compile a broad and diverse collection of 38 high-quality articles. By covering a wide range of sports, detection techniques, and publication years, this review offers a more comprehensive and up-to-date perspective.

In conclusion, after researching and screening the articles, an in-depth analysis of each individual article was conducted and compiled into the next section. The next chapter presents various statistics on the gathered articles through easy-to-read graphs and ratios, as well as some initial insights into what to expect from these results. The section provides a summary of each article and concludes with an organized table that lists the main points of each study.

Results

This chapter presents the research results and provides various statistics on these findings. As explained above, the search yielded 38 relevant studies for this review. As for the sport each study focuses on, 12 (31.58%) studies are centered around football/soccer, five (13.16%) in tennis, three (7.89%) articles are on basketball and table tennis, two (5.26%) articles focus on boccia, golf, handball, and volleyball each, and 1 (2.63%) article refers to padel, badminton and rugby each, the latter being the only sport with a ball that is not traditionally round. There are also three (7.89%) articles that do not pertain to any specific sport and instead research object detection applied to sports in general. Of all the sports-related articles in this review, seven (18.42%) focus solely on detecting humans. However, their contributions are still considered valuable to this analysis. At last, there is also one (2.63%) article that researches ball detection but is not sports-related, although it is a use-case that the authors mention.

These statistics can be visualized in Fig. 2. There is a clear bias towards football (and, to some extent, tennis), with the highest number of relevant studies found. This bias is due to research availability and is linked to the sport’s popularity. Football is the most popular sport in the world, and it shows. It is also possible that these sports can simply take advantage of computer vision more effectively than others, especially in detecting sports balls or players. This bias may skewer results, such as showing increased popularity and use of object detectors that are often used in football studies and systems, for example. Hence, it is recommended to take some of the following statistics with some caution.

Figure 2 Number of studies gathered, categorized by sport.

Regarding the type of detectors, five (13.16%) articles solely research machine learning methods or traditional computer vision processing techniques, bearing no touch on deep learning-based methods, which in recent years have grown in popularity and shown capable of achieving greater accuracy in more complex tasks such as image classification and semantic segmentation. Of the deep learning-based methods used, a staggering 26 (68.42%) articles either employed some version of YOLO or extensively compared their developed algorithms to some version of YOLO as a benchmark. It demonstrates the increasing prevalence of this technique over the years since its debut in 2015. Also very popular, eight (21.05%) articles either extensively compared or used an established form of R-CNN, be this Faster R-CNN with five (13.16%) articles or Mask R-CNN with three (7.89%) articles. The SSD method also had a significant presence, being used in five (13.16%) of the studies. The authors of three (7.89%) articles self-proposed an enhanced CNN-based algorithm, and the RetinaNet method was used once (2.63%). The popularity of these base methods can also be analysed in Fig. 3. Note that a single article can adhere to multiple detection methods, whether by using, testing, or extensively reviewing, which is why the graph holds different values compared to the above statistics.

Figure 3 Number of studies collected, categorized by detection method.

Deep learning methods can be further categorized according to the number of stages. As the name suggests, two-stage methods split object detection into two sequential stages, while one-stage methods bypass the need for region proposal. Table 2 aggregates every type of deep learning-based object detector researched throughout these works, divided by detector type.

Table 2 Examples of object detectors by type: one-stage vs two-stage.

Detector type	Detector base	Detector version	
One-stage	YOLO	YOLO (Deepa et al., 2019; Decorte et al., 2024; Solberg et al., 2024)	
		YOLO-tiny (Calado et al., 2019b)	
		YOLOv2 (Burić, Pobar & Ivasic-Kos, 2018a, 2018b)	
		YOLOv3 (Tian, Zhang & Zhang, 2020; Zhang et al., 2020; Hiemann et al., 2021; Meneghetti et al., 2021; Li & Zhao, 2024; Modi et al., 2024; Li et al., 2023; Liu et al., 2021)	
		YOLOv3-tiny (Zhang et al., 2020; Sheng et al., 2020; Meneghetti et al., 2021)	
		YOLOv4 (Meneghetti et al., 2021; Li & Zhao, 2024; Kulkarni et al., 2023)	
		YOLOv4-tiny (Meneghetti et al., 2021)	
		YOLOv5 (Modi et al., 2024; Zhao, 2024; Li & Zhao, 2024)	
		YOLOv7 (Hassan, Karungaru & Terada, 2023)	
		YOLOv8 (Modi et al., 2024; Esfandiarpour, Mirshabani & Miandoab, 2024; Luo, Quan & Liu, 2024; Fu, Chen & Song, 2024; Hu et al., 2024; Yin et al., 2024)	
		xYOLO (Barry et al., 2019)	
		YOLO-HGNet (Yang et al., 2024b)	
		YOLO-T2LSTM (Li, Luo & Islam, 2024)	
One-stage	SSD	SSD (Li & Zhao, 2024; Deepa et al., 2019)	
		SSD-MobileNetV1 (Renolfi de Oliveira et al., 2019; Pawar et al., 2021)	
		SSD-MobileNetV2 (Meneghetti et al., 2021)	
		SSD-MobileNetV3 (Meneghetti et al., 2021)	
One-stage	RetinaNet	RetinaNet (Fujimoto et al., 2024)	
Two-stage	Faster R-CNN	Faster R-CNN (Deepa et al., 2019) (Zhang et al., 2020; Keča et al., 2023; Fujimoto et al., 2024; Li & Zhao, 2024)	
Two-stage	Mask R-CNN	Mask R-CNN (Burić, Pobar & Ivasic-Kos, 2018a, 2018b; Fujimoto et al., 2024)	
Two-stage	Other CNN	CNN (Reno et al., 2018; Teimouri, Delavaran & Rezaei, 2019; Li, Luo & Islam, 2024)	

This aggregation highlights the popularity of YOLO-based algorithms and the numerous sub-versions of YOLO, demonstrating their high versatility and adaptability. Many researchers also opt for a hybrid system, combining YOLO with other techniques to strengthen its weaknesses or build upon its already strong characteristics.

Taking an individual look at each base detector, Faster R-CNN has been a staple of object detection since its debut. While the original R-CNN model achieved desirable object detection results, it had some significant shortcomings, particularly in terms of speed. Faster methods were introduced, starting with Fast R-CNN, which was quickly followed by Faster R-CNN, one of the best versions of the R-CNN family. This method replaced a selective search algorithm to compute region proposals with a superior region proposal network (RPN) Zengin et al. (2020). Although this was a significant improvement over its predecessors, the delay in prediction is still its biggest limitation. However, it remains a good choice when object detection is done offline, and inference speed is not the highest priority. For more details on Faster R-CNN, refer to Ren et al. (2017).

As a later advancement to the Fast R-CNN, Mask R-CNN added a branch for predicting an object mask in parallel with the existing bounding box recognition. Utilizing pixel-to-pixel alignment, this state-of-the-art algorithm in image and instance segmentation achieves significant improvements in terms of accuracy and precision. It adds a small overhead to Faster R-CNN, inheriting its major weakness of high computation cost. For the more interested reader on the inner works of Mask R-CNN, refer to He et al. (2017).

The single-shot detector for multi-box predictions is one of the fastest methods for achieving real-time computation of object detection tasks. While the two-stage algorithms above can achieve high prediction accuracies, SSD removes the RPN and instead uses multiscale features and default boxes to improve its inference performance. The main limitations of this method are the decrease in image resolution to lower quality and the difficulty in detecting small-scale objects. For more details about SSD, refer to Liu et al. (2016).

The RetinaNet uses SSD detection capabilities to achieve superior results in accuracy-related metrics. This model is built upon the feature pyramid network (FPN) and utilizes focal loss to address some of the issues presented by the previous SSD algorithm. The result is a well-balanced algorithm that is not as fast as YOLO nor as accurate as Faster R-CNN, but it provides a safe choice for a wide array of applications. For more details on RetinaNet, the authors refer to Lin et al. (2017).

YOLO became one of the most popular models due to its neural network architecture, which enables high computation and processing speeds, especially in real-time, compared to most other training and detection methods, while providing an overall acceptable accuracy rate. This architecture allows the model to learn and develop an understanding of numerous objects more efficiently. However, it does not have all the advantages, as it fails to detect smaller objects accurately due to its lower Recall rate. Detecting multiple objects that are close to each other can also be challenging due to the limitations of bounding boxes. For more details on YOLO, refer to Redmon et al. (2016).

Table 3 summarizes how each of the assessed models performs in terms of inference speed and detection accuracy in an overall manner.

Table 3 Strengths and weaknesses of the various object detectors.

Detector	Speed	Accuracy	
Faster R-CNN	Low	High	
Mask R-CNN	Very low to low	High to very high	
SSD	High	Medium	
RetinaNet	Medium to high	Medium to high	
YOLO	Very high	Low	

In general, and depending on algorithm enhancements and hardware available, YOLO and SSD are preferable solutions when real-time inference times are a necessity. In contrast, two-stage algorithms take precedence when higher precision and accuracy are of the utmost priority.

To evaluate all the different models and algorithms, various evaluation metrics (PaperspaceMetrics, 2019) are used, most of which are based upon the basic concepts of true positives (TP, correct detection—both the prediction and true value were positive), false positives (FP, incorrect detection—prediction was positive, but true value was negative), false negatives (FN, ground truth not detected—prediction was negative, but true value was positive) and true negatives (TN, correct misdetection—both the prediction and true value were negative). One of these metrics is precision, the ability of a model to identify only the relevant objects. It is the percentage of correct predictions, given by Eq. (1):

(1) Precision=TPTP+FP.

Another metric is recall (or detection rate), the ability of a model to find all relevant cases, given by Eq. (2):

(2) Recall=TPTP+FN.

F1-score is the weighted average of precision and recall, as given by Eq. (3):

(3) F1=2∗TP2∗TP+FP+FN.

Average precision (AP) and mean average precision (mAP) are also widely used to assess a detection model. In essence, AP is a weighted sum of the precision values, where the weights are the increase in recall between consecutive points on the precision-recall curve. Considering n is the index of a point on the precision-recall curve, a graph that visualizes the trade-off between precision and recall of a model for different confidence thresholds, Pn is the precision at point n, Rn is the recall at point n, and R0 is assumed to be 0, the formula for AP is given by Eq. (4):

(4) AP=∑n(Rn−Rn−1)∗Pn.

The mAP is calculated by averaging AP values across all classes. Put simply, considering n the number of classes, mAP is given by Eq. (5):

(5) mAP=∑(AP)n.

Accuracy measures the overall correctness of a model’s predictions, encompassing all four basic concepts of TP, TN, FP, and FN, as given by Eq. (6):

(6) Accuracy=TP+TNTP+TN+FP+FN.

Other metrics include the inference time (or inference speed), which corresponds to the calculation time for each frame in milliseconds and is critical to evaluate a model’s computational cost and real-time application, and Intersection over Union (IoU), which measures the overlap between the predicted and ground truth bounding boxes to determine if a predicted object is a TP or an FP. In sum, IoU is given by the area of overlap by the area of union between bounding boxes, mathematically written as in Eq. (7):

(7) IoU=A∪BA∩B.

From the 38 articles evaluated, as depicted in Fig. 4, the most used metric was Inference Time, with 23 appearances (60.53%), revealing that real-time application (or, at least, the system’s computational cost) is a major concern when applying object detection to sports. The popularity of the remaining metrics is as follows: Accuracy with 20 (52.63%), precision and recall with 18 (47.37%), F1-score with 12 (31.58%), mAP with 10 (26.32%), AP with 6 (15.79%), and finally, IoU with 5 (13.16%) appearances.

Figure 4 Distribution of studies by evaluation metric.

The following subsections present brief reviews of the articles gathered, organized by sport, and chronologically. The studies focused on football are presented first, as they represent the majority of sports, followed by tennis. After that, the reviews on the remaining studies are presented, given the minimal number of studies on each of these remaining sports.

Football focused studies

This subsection presents the studies that focus on object detection applied to football, organised by ascending year of publication.

The study undertaken by Teimouri, Delavaran & Rezaei (2019) provides another real-time ball identification system for football-playing robots. The suggested solution leverages a two-step approach: a region proposal algorithm to discover prospective ball regions, followed by a lightweight CNN to refine and categorise ball candidates. The technique was tested using datasets from RoboCup (Visser et al., 1997) contests, including unseen ball patterns and different lighting conditions, to test generalizability and resilience. The proposed pipeline managed great overall results on precision and recall. Major challenges were variable lighting, motion blur, and recognizing balls with new patterns.

Renolfi de Oliveira et al. (2019) also addresses the challenge of humanoid football robots under constrained hardware constraints for ball detection. Aiming to optimize performance on CPU-only systems, the authors tested several pre-trained configurations of the MobileNet detector on a custom dataset of images of football balls under various conditions. The training used the ImageNet (Deng et al., 2009) subset, specifically the ILSVRC (Russakovsky et al., 2015) subset. The ideal configuration, MobileNet, with a 0.75 width multiplier and 128-pixel resolution, achieved acceptable performance with balanced accuracy, indicating appropriateness for the quick, CPU-only needs of sports robotics. Nonetheless, reduced network parameters impaired detection accuracy, showing a trade-off between speed and precision.

Barry et al. (2019) provides a model built for real-time object identification on low-end hardware, notably the Raspberry Pi 3 Model B, equipped on a mobile robot. The study alters the YOLOv3-tiny model to develop xYOLO, optimized with reduced input size, fewer filters, and selective usage of XNOR layers. The proposed model is experimented with using Darknet (Redmon, 2013) on a custom dataset. xYOLO achieves significant computational benefits while trading some Accuracy to match the speed requirement of the robot. According to the authors and their results, xYOLO is a breakthrough in computational efficiency, making it suitable for real-time and embedded devices. Tests conducted on xYOLO showed that it exceeded already lightweight models, such as Tiny-YOLO, in speed, making it a feasible candidate for cases where quick decisions are vital. However, the results also displayed subpar accuracy ratios.

Fatekha, Dewantara & Oktavianto (2021) attempts to enhance the object detection skills of wheeled goalkeeper robots deployed in football. Motivated by the need to identify footballs while minimizing false detections reliably, the researchers devised a color-based segmentation system that separates in-field objects from those outside. Using HSV colour space conversion and morphological procedures, the system detects field boundaries and the ball based on their distinct colour features. This enables the system to filter out locations beyond the field and accurately identify the ball’s positions. Video footage collected during the field activity was used to test the system, with measurements centred on processing time (average of 41.6 ms per frame) and frame rate (31 fps), which proved adequate for real-time applications. However, non-field items with similar hues to the ball could still cause occasional false detections.

Meneghetti et al. (2021) evaluates the performance of several convolutional neural network architectures, including MobileNet (versions 2 and 3) and YOLO (versions 3 and 4, along with YOLO-tiny variants), for football ball detection on central processing unit (CPU) only embedded systems, a common setup in mobile robots. With the models pre-trained using the Common Objects in Context (COCO) (Lin et al., 2014) dataset, results showed that MobileNetV3 models achieved a better balance of speed and accuracy under limited CPU conditions, whereas YOLO models, optimized for graphics processing unit (GPU) processing, exhibited prohibitive inference speeds. The authors also offer important benchmarks for real-time applications in autonomous robots.

The study by Hassan, Karungaru & Terada (2023) focuses on recognizing handball events in football, wanting to increase decision-making accuracy in recognizing such incidents. The system featured a single-camera setup and utilized a combination of YOLOv7 for object identification, Detectron2 for instance segmentation, a Kalman filter for object tracking, and the Separating Axis Theorem (SAT) as a robust method for detecting overlaps between hands and balls. The dataset consisted of images of hands and balls, annotated using the COCO format. Metrics like detection accuracy and confusion matrices confirmed the method’s effectiveness. However, there were still issues with false positives and occlusion.

The work by Li et al. (2023) introduced a real-time football detection system designed for immersive live sports broadcasting. Using 36 monocular cameras arranged around a stadium, the system captures high-resolution video feeds processed by GPU-accelerated servers. The optimized detection architecture, based on YOLOv3, is capable of identifying very small or distant targets in challenging conditions, such as occlusion and lighting variation. Single-camera detections are combined through bundle adjustment to reconstruct the ball’s position in 3D space. Evaluation metrics focused on accuracy, inference time, and generalizability, with system testing confirming its effectiveness across various sports, including basketball and rugby. The study addresses common limitations in 3D small-object detection, such as computational cost and real-time constraints. The authors emphasize the potential of this system in enhancing viewer experience through AI-driven immersive media, such as augmented and virtual reality integrations. A key contribution was the integration of deep learning with 3D vision. However, limitations persist in maintaining Accuracy under complex environmental changes, ensuring reliable camera calibration, and the high cost of such a system.

Zhao (2024) proposes an enhanced version of the YOLOv5 object identification model designed for a football player and ball detection in real-time scenarios. This model was enhanced with lightweight multi-scale feature extraction modules, reduced computational complexity, and more precise bounding box prediction. These modules were Simplified Spatial Pyramid Pooling Fast (SimSPPF), GhostNet, and slim scale detection. It achieved high performance on metrics like precision, recall, and mAP50, with a 45% reduction in computational costs and a 53% smaller model size compared to the baseline YOLOv5. The biggest problem that persisted was obscurity between players.

Modi et al. (2024) presents a study that provides an object tracking system that combines YOLOv8 with optical flow analysis to track a football in real-time. Pre-processing and noise reduction techniques, like Gaussian blur and non-maximum suppression (NMS), were applied to improve accuracy. The DFL Soccer Ball Detection dataset (Mortenson, 2022) was used for model training. Key measures, such as true positives (TP), false positives (FP), and false negatives (FN), were used to evaluate its performance, and results suggest that YOLOv8 surpassed earlier YOLO models, notably YOLOv3 and YOLOv5, in terms of both accuracy and efficiency. The lightweight architecture of YOLOv8 enables it to be implemented on computationally restricted devices, making it well-suited for real-time and embedded devices.

The study by Esfandiarpour, Mirshabani & Miandoab (2024) explores and enhances ball identification algorithms for sports robots, notably for football applications. The researchers examine four common methods: colour detection, Circle Hough Transform (CHT), frame differencing, and YOLOv8. Using a custom dataset of seven films with varying frame rates, ball sizes, and colours, they scored each system based on accuracy, noise sensitivity, and processing time. Although YOLOv8 achieved the highest standalone accuracy (96.55%), hybrid models that combine colour detection with either CHT or YOLOv8 showed significant improvements in detection speed and accuracy, making the latter hybrid especially suitable for real-time applications.

The work by Fu, Chen & Song (2024) presents an enhanced detection model named HPDR-YOLOv8, tailored for football, where detecting distant, occluded, and deformable objects like players and balls is challenging. According to the authors, this model, based on YOLOv8n, introduces several innovations, such as: a neck for improved feature fusion, a detection layer for small object enhancement, and a re-parameterized detection head for reduced computation and improved efficiency. Evaluations on the SoccerDB (Jiang et al., 2020) and custom datasets showed improvements in mAP, precision, and recall over the baseline YOLOv8n. This work presents a good example of how a hybrid system can improve on the weaknesses of a base model. The authors highlight its potential for real-time applications in football analytics.

The article by Solberg et al. (2024) introduces a novel framework, PlayerTV, that automates the creation of player-specific football highlight videos. The system combines object detection based on YOLO models, tracking, optical character recognition (OCR), and color analysis to identify players based on their team numbers and colors. PlayerTV also features a user-friendly interface that allows users to configure and extract highlight clips of individual players. Evaluated on a custom dataset, the framework demonstrated strong team identification accuracy but struggled with specific player identification due to OCR limitations. The system was tested on a GPU-equipped computing cluster and is open-sourced for replication and further development.

Tennis focused studies

This subsection presents the studies that focus on object detection applied to tennis, arranged chronologically by their publication date.

The work by Ali Shah et al. (2018) focuses on constructing a real-time tennis ball detection and tracking system utilising image processing. Motivated by the difficulty in consistently monitoring a fast-moving object in a complicated background, the article provides a simplified approach based on contour and centroid detection methods. The video is streamed from an IP camera and processed on a laptop, which communicates with a robot equipped with a low-powered embedded CPU. Even with synchronization difficulties between the camera and motor, negatively affecting tracking precision, the model still achieved great detection and tracking rates.

Reno et al. (2018) offers a CNN for real-time detection of tennis balls, tackling the constraints of standard tracking approaches in sports. Using a custom dataset of pictures from genuine tennis matches and sessions, their CNN model acts on individual picture patches, obtaining great accuracy (98.77%) by discriminating between “Ball” and “No Ball” patches without relying on background subtraction or frame differencing. Even though the model proved efficient, false positives, hazy balls, and court lines were still challenges.

Deepa et al. (2019) evaluates three object recognition algorithms for real-time tennis ball tracking: YOLO, SSD, and Faster R-CNN. Some of the work was already introduced in the 2nd section of this review. The models were trained on bespoke video frames. SSD outperformed YOLO and Faster R-CNN due to its Accuracy and processing efficiency. However, the system still suffers from occlusions and quick ball motions.

The work by Tian, Zhang & Zhang (2020) focuses on identifying small, fast-moving objects, like tennis balls. The authors apply deep CNNs tailored for tiny object detection and compare them to YOLOv3, obtaining great performance results. The dataset used was a combination of broadcast video of tennis competitions and the sports ball class of the MS-COCO dataset. The key contributions include designing a framework that efficiently detects small objects in real-world sports scenarios and offering enhancements for practical implementation.

Li & Zhao (2024) proposes an upgraded version of YOLOv5 for real-time tennis ball recognition, tackling the problems of complex scenes. The method incorporates three main enhancements: a refined loss function, the addition of a convolutional attention mechanism, and a new feature extraction network. Experiments were conducted on a desktop machine, utilizing a bespoke image dataset with varying lighting and court settings to mimic real-world applications. Results show that this model easily reached real-time inference speed with a model size of 12.1 MB and competing mAP values.

Studies on other sports

This section outlines research efforts exploring object detection in less common sports, ordered chronologically by publication year.

Authored by Li et al. (2015), this article offers a universal shape-based detection method to recognise ball-shaped objects using CHT and an R-CNN framework for classification. Shape detection was evaluated on recall, the number of proposal bounding boxes per image, and the best average overlap. Regarding object detection, utilizing the ILSVRC dataset and focusing on eight-ball classes (baseball, basketball, croquet, football, golf, ping-pong, tennis, and volleyball), the authors compared their proposed model with two other methods: Binarized Normed Gradients (BING) and Selective Search. Results also showed that the proposed method achieved the best average overlap of the three models and strong recall.

Burić, Pobar & Ivasic-Kos (2018a, 2018b) have authored two similar works to recognize actions in handball scenes. The authors present an overview of two deep learning-based algorithms, YOLOv2 and Mask R-CNN, and assess their performance using a bespoke dataset comprising handball match recordings, trained on the MS-COCO dataset. These deep learning approaches are also pitted against a conventional technique, MOG, which performed worse in almost every aspect. Through typical detection measures, such as precision and F1-score, Mask R-CNN came out ahead. YOLO, however, had higher recall and demanded much fewer processing resources. The study reveals the trade-offs between speed and accuracy in object detection for real-time sports analysis, and that training on custom datasets significantly enhances both models’ performance.

The article by Calado et al. (2019a) analyses the potential of boccia to stimulate physical activity among elderly adults. Using a low-cost webcam and a Python-based object detection algorithm, the researchers constructed a real-time system that identifies and tracks boccia balls by colour to compute scores. The method was proven to be accurate in 19/20 instances, tested on recordings of simulated games. While the system operated effectively, the dataset used was quite small, and limitations like sensitivity to lighting conditions, the camera’s narrow field of view (FOV), and the necessity for manual adjustments.

In another article by Calado et al. (2019b), the authors build upon the real-time ball identification model previously constructed, aiming to enhance boccia game analysis. Two detection methods were compared: HOG-SVM and Tiny-YOLO. Both models performed comparably with over 90% accuracy in offline and real-time tests. However, Tiny-YOLO demonstrated somewhat superior precision and frame rates. The custom dataset comprises photos from various boccia game scenarios, providing flexible detection in diverse environments. Challenges remain, such as difficulty detecting balls on court lines of similar colors and maintaining detection dependability necessary for scoring.

Wang et al. (2019) offers a high-speed stereo vision system for tracking golf ball motion, focused on indoor golf simulations. Different types of hardware were considered for this project, including infrared (IR) sensors, radar, and high-speed stereo cameras. The IR approach was low-cost but had a limited detection range and detection rate. The radar approach provided an improved detection rate but struggled to detect low-speed movements and accurately measure ball spin. The authors used two high-speed cameras running at 810 frames per second (FPS) and an embedded device to measure critical motion metrics such as speed, trajectory, and spin without the need for strobe lights. The suggested system uses dynamic region-of-interest (ROI) algorithms for hitting detection and real-time ball position tracking. Custom-designed markers on the golf ball boost spin measurement. At the same time, a P-tile approach with non-ball object filtering was utilized to address issues of low image resolution and underexposure. The system exceeded a commercially recognized device, the GC2, in terms of speed and accuracy, tested on 500 shots under indoor lighting.

The study by Zhang et al. (2020) aims to develop a real-time system for detecting golf balls using CNNs, combined with a discrete Kalman filter for tracking, while addressing the challenges of motion blur, small object size, and high-speed movement. The authors trained three detection models, Faster R-CNN, YOLOv3, and YOLOv3-tiny, on a custom-tagged dataset of golf ball images and video sequences. The authors’ results showed that Faster R-CNN had the best accuracy. However, YOLOv3-tiny excels in speed while still maintaining competitive accuracy, making it preferable for real-time applications where a high frame rate is key.

Sheng et al. (2020) addresses the difficulty of real-time small object identification, specifically for monitoring table tennis balls. The researchers proposed an optimized version of the YOLO3-tiny model. The solution utilized network pruning to reduce redundancy and a novel feature fusion mechanism to boost detection accuracy for small objects. The approach achieved an mAP of 0.971 and a detection speed of 3.4 ms per image on a medium-end GPU evaluated on a custom-built dataset. The authors highlight the superiority of their proposed model to a baseline approach, in terms of balance between speed and Accuracy.

The study by Hiemann et al. (2021) intended to develop a universal real-time sports ball detection system, focusing on accurately tracking fast-moving, small objects such as volleyballs. The researchers upgraded the YOLOv3 model to boost speed and accuracy through key enhancements, including multi-resolution feature extraction, the inclusion of motion information, and data augmentation targeted at improving generality and precision in sports videos. Pre-trained on COCO and tested on a bespoke dataset from volleyball matches, the model achieved satisfactory results in metrics such as precision, recall, and F1-score, while delivering minimal latency on a medium-to-high-end GPU. While the results revealed enhanced Accuracy and speed adequate for real-time applications, the study found that the dataset used for training was limiting the system’s capabilities when applied in other sports.

The article by Balaji, Karthikeyan & Manikandan (2021) examines the application of metaheuristic algorithms to enhance object detection in volleyball players, with the primary objective of addressing challenges such as occlusion, motion blur, and variations in color, lighting, and background. The study examines and compares three algorithms: Firefly, teaching-learning-based optimization (TLBO), and cuckoo search algorithm. The latter displayed great performance, delivering high accuracy and precision in spotting players among quick movements and background disturbances. However, occlusion remains a significant problem.

Pawar et al. (2021) focuses on constructing an effective object recognition and tracking system for holonomic robots, employing the SSD-MobileNet architecture, designed for embedded systems. Through fine-tuning on a proprietary dataset of rugby ball images, the authors developed a model capable of accurately identifying and tracking objects with minimal processing requirements, suitable for deployment on limited hardware such as the Raspberry Pi 4 Model B and Arduino Mega. The base model and the 8-bit quantized model were compared. While the tuned model still did not reach real-time inference speeds, it achieved a significant relative increase in FPS while maintaining comparable accuracy.

The article by Liu et al. (2021) demonstrates an object detection framework tailored for sports analytics, particularly in crowded scenarios such as those found in football or basketball. The authors propose a method that detects and matches related objects, such as a player and their equipment, simultaneously from a single proposal box using an implicit association mechanism. This approach avoids the shortcomings of traditional detection pipelines, which treat related objects independently and rely on post-hoc matching, often failing in crowded scenes. The method is based on YOLOv3, enhanced with a Feature Pyramid Network (FPN), and evaluated using both custom and public datasets, including the COCO dataset. Detection evaluation metrics include accuracy, AP, IoU, and inference time. Experiments demonstrate significant performance gains in both detection and matching tasks. While the method shows performance drops for related objects with less spatial overlap, it excels in high-overlap scenarios.

Keča et al. (2023) is intended to boost ball detection in robotics competitions using deep learning deployed on an educational robot based on a Raspberry Pi. The authors offer a bespoke dataset of aluminium-wrapped ball images under diverse lighting conditions, aiming to generalize the model for real-world situations. They explored the possibility of using a camera, HT, and deep learning methods, such as CNNs, for object detection and U-Net-based semantic segmentation. The authors extensively tested many different networks, which, by themselves, cannot be considered a detection model, but together with an algorithm such as Faster R-CNN, as was the case, can. Most notoriously, when it came to object detectors, the fastest object detector was MOBILENET_V3_LARGE_320, and RESNET50 was the most accurate. Regarding semantic segmentation, MOBILENET_V2 proved the quickest and EFFICIENTNET-B0 the most accurate. There is no single best method, and a trade-off always exists between precision and frame rate. Key parameters used in the testing phase included precision, recall, DICE coefficient, IoU, model size, and temporal measurements for real-time application.

The study by Kulkarni et al. (2023) explored a novel approach to detecting and recognizing table tennis strokes using only ball trajectory data, thereby eliminating the need for player-focused video or wearable sensors. The authors propose a streamlined, non-intrusive detection technique using 3D ball position data captured via a 60-camera optical tracking system. Ball detection was tested on two deep learning methods, YOLOv4 and TrackNetv2, on a custom dataset and assessed through accuracy, precision, and F1-score. While the method offers a practical and data-efficient alternative to video-based techniques, its reliance on high-quality trajectory data limits broader applicability. Overall, the study presents a promising step toward applying machine learning in sports analytics, particularly in enhancing coaching in table tennis.

The article by Decorte et al. (2024) introduces a multi-modal algorithm for hit detection and positioning analysis in padel, utilizing audio and video extraction features. The authors use audio to identify ball hits and perform ball detection from a video feed only around the time of this recognition. This approach helps tackle age-old challenges like occlusion. Still, it creates new difficulties, like noise, such as people shouting, and different sounds relative to how the ball was hit (angle of the racket, velocity of the ball and racket, ball’s travelling direction, etc.). The hit detection model is a modified version of sound event detection (SED) using spatial features and recurrent neural networks (RNN). When it comes to pose, and player tracking, YOLO is used. The proposed algorithm showed satisfying results, achieving an average F1-score of 92%, in between other metrics tested. The authors also make available a dataset of annotated images from padel rallies.

Fujimoto et al. (2024) address the enhancement of accuracy for table tennis ball detection by fine-tuning pre-trained (using COCO) object detection algorithms, addressing shortcomings with existing models like high false-positive rates. Utilizing Google Colab, they deployed three pre-trained models: Mask R-CNN, Faster R-CNN, and RetinaNet. The models’ precision and inference time were tested, showing a significant increase in both metrics. Mask R-CNN obtained the biggest improvements. This article shows the importance of enhancing detectors through relevant training and fine-tuning for a specific task. However, this may decrease the generalizability of the detector in different scenarios.

The study by Luo, Quan & Liu (2024) addresses the difficulty of accurately recognizing small, fast-moving sports objects in complex backgrounds. Leveraging YOLOv8, the authors introduce a small object detection head and a Squeeze-and-Excitation (SE) attention mechanism, specifically tailored to capture smaller, fast-moving objects. These advancements refine the model’s precision and efficiency, providing real-time tracking. With data augmentation using the Mosaic approach and training, the model exhibited improved performance metrics compared to the original YOLOv8.

The article by Li, Luo & Islam (2024) proposes a hybrid YOLO-T2LSTM model to enhance basketball player detection and movement recognition. This model incorporates the YOLO algorithm for player detection and integrates a long short-term memory (LSTM) and type-2 fuzzy logic for action classification, utilizing a multi-feature extraction approach based on VGG16, VGG19, and ResNet50 backbones. A median filter was also integrated for noise reduction. The proposed model was compared to various advanced feature extraction methods to assess its performance, which was successful, with an average recognition rate of 99.3% over 8 different actions.

The article by Yang et al. (2024b) introduces YOLO-HGNet, an innovative model that enhances the detection and classification of badminton strokes. The authors introduced HourGlassNetwork (HGNet) as the backbone network of the model, based on YOLOv8, to enhance the detection efficiency and overall accuracy. Then, they improved the model’s data and feature processing capabilities by integrating a new attention mechanism, ACmix, and Depth-Wise Convolution (DWConv). The authors also use focal modulation to emphasize key regions in the image further, improving action recognizability. Extensive experiments were conducted using various combinations of networks, models, and features. The proposed hybrid model showed great results, taking the lead in accuracy.

The study by Hu et al. (2024) presents a novel online tracking method tailored for basketball, addressing the challenges of complex multi-object occlusion commonly encountered in team-based sports. The proposed model integrates projected position tracking using appearance and motion features to improve trajectory association. The authors employ YOLOv8 for detection, with all experiments conducted on a high-end machine and assessing results using the TeamTrack dataset (Scott et al., 2024). The study’s main contributions lie in its domain-specific enhancements and robust handling of occlusions.

This work by Yin et al. (2024) proposes a novel approach to multi-object tracking in basketball, referred to as Sports-vmTracking, which addresses the challenges posed by frequent occlusions and the visual similarity of players. The method is benchmarked on a custom dataset, compared against a baseline YOLOv8 model, and evaluated using Accuracy and comprehensive multi-object tracking metrics. Through a single camera, the authors encountered difficulties in dynamic environments due to occlusion. Nonetheless, great results were obtained, and the authors aimed to explore multi-camera setups to further reduce the challenge faced by occlusion.

Summary of researched studies

In this subsection, the research articles reviewed are aggregated in Table 4, along with their year of publication, purpose, target objects, detection methods used, evaluation metrics used, and dominant challenges felt.

Table 4 Listing and summarization of studies.

Study	Purpose	Target	Detection methods	Evaluation metrics	Challenges	
Li et al. (2015)	To describe a novel approach for generating object proposals for ball detection	Sports balls	R-CNN, contouring, CHT,	mAP, Recall, Inference time	Small or distant objects, computational cost	
Ali Shah et al. (2018)	To achieve an embedded approach for ball detection and tracking using image processing	Sports balls	Colour detection, Gaussian blur, contouring, centroid detection	Accuracy	Complex backgrounds, motion blur	
Reno et al. (2018)	To present an innovative deep learning approach to tennis balls identification	Sports balls	CNN	Accuracy, Precision, Recall	Motion blur, high number of false positives	
Burić, Pobar & Ivasic-Kos (2018a)	To compare YOLO and Mark R-CNN for handball detection in real-world conditions	Sports balls and humans	YOLOv2, Mask R-CNN	Precision, Recall, F1-score	Occlusion, computational cost, high number of false positives	
Burić, Pobar & Ivašić-Kos (2018b)	To provide an overview of CNN detection methods for handball analysis	Sports balls and humans	YOLO, Mask R-CNN, MOG	Precision, Recall, F1-score	Occlusion, lighting, computational cost	
Teimouri, Delavaran & Rezaei (2019)	To propose a low-cost ball detection method for football robots	Sports balls	CNN	Accuracy, Precision, Recall, Inference time	Lighting, motion blur, underfitting	
Renolfi de Oliveira et al. (2019)	To investigate the performance of a vision system for object detection under constrained hardware, trained on a football dataset	Sports balls	SSD-MobileNet	Accuracy, Precision, mAP, Recall, F1-score, Inference time	Accuracy trade-off, computational cost	
Barry et al. (2019)	To develop a lightweight real-time model (xYOLO)	Sports balls and goalposts	xYOLO	mAP, F1-score, Inference time	Accuracy trade-off	
Deepa et al. (2019)	To describe a systemic approach for trajectory estimation of tennis balls	Sports balls	YOLO, SSD, Faster R-CNN	Accuracy, Inference time	Occlusion, motion blur	
Calado et al. (2019a)	To propose a ball-detection system for boccia to motivate elderly physical activity	Sports balls	Contouring, centroid detection, colour detection	Accuracy	Lighting, underfitting	
Calado et al. (2019b)	To have a versatile algorithm for boccia balls detection	Sports balls	Tiny-YOLO, HOG-SVM	Precision, AP, Recall, Inference time	Computational cost, high number of false negatives	
Wang et al. (2019)	To present a high-speed stereo vision golf ball tracking system	Sports balls	ROI, P-tile, noise filtering	Accuracy, Recall, Inference time	Occlusion	
Tian, Zhang & Zhang (2020)	To propose an anchor-free tennis balls object detector	Sports balls	YOLOv3	Accuracy, Precision, Recall, F1-score	Motion blur, small or distant objects	
Zhang et al. (2020)	To propose an efficient solution for real-time golf ball detection and tracking	Sports balls	YOLOv3, YOLOv3-tiny, Faster R-CNN, Kalman filter	Precision, mAP, Inference time	Small or distant objects, underfitting	
Sheng et al. (2020)	To propose a real-time one-stage algorithm based on feature fusion for table tennis balls detection	Sports balls	YOLOv3-tiny	mAP, Inference time	Underfitting	
Fatekha, Dewantara & Oktavianto (2021)	To enhance detection algorithms for sports robotics in football using colour-based segmentation	Sports balls	Colour-based segmentation, morphological operations, contouring	Inference time	High number of false positives	
Meneghetti et al. (2021)	To evaluate the performance of detection systems in constrained hardware	Sports balls	YOLO (v3, v4, tiny-v3 and tiny-v4), SSD (v2 and v3)	AP, Inference time	Computational cost	
Hiemann et al. (2021)	To address the detection of small and fast-moving balls in sports in real-time	Sports balls	YOLOv3	Precision, AP, Recall, F1-score, IoU, Inference time	Underfitting, motion blur	
Balaji, Karthikeyan & Manikandan (2021)	To research volleyball player detection using innovative Metaheuristic algorithms	Humans	Firefly, TLBO, Cuckoo Search	Accuracy, Precision, Recall	Occlusion	
Pawar et al. (2021)	To create a robust tracking algorithm based on a custom rugby dataset to replicate industrial object detection	Sports balls	SSD-MobileNet	Accuracy, Inference Time	Underfitting, computational cost	
Liu et al. (2021)	To propose a method that detect and matches players to their equipment with a single bounding box	Humans	YOLO, FPN	Accuracy, AP, IoU, Inference time	Occlusion, Overlap of bounding boxes	
Hassan, Karungaru & Terada (2023)	To propose a method to detect handball events on football games	Sports balls and humans	YOLOv7, instance segmentation, Kalman filter, separating axis theorem	Accuracy	Occlusion, High number of false positives	
Keča et al. (2023)	To explore the combination of HT and deep learning methods for ball detection on a low-powered device	Sports balls	Faster R-CNN	Precision, Recall, F1-score, IoU, Inference time	Computational cost	
Li et al. (2023)	To improve immersiveness in live sports broadcasting via real-time 3D detection	Sports balls and humans	YOLOv3	Accuracy, Precision, Recall, IoU, Inference time	Complex backgrounds	
Kulkarni et al. (2023)	To detect and identify table tennis strokes using ball trajectory data, eliminating the need for player-focused video and wearable sensors	Sports balls	YOLOv4, TrackNetv2	Accuracy, Precision, F1-score, Inference time	Accuracy	
Zhao (2024)	To propose an improved version of YOLOv5 for football and player recognition	Sports balls and humans	YOLOv5	Precision, AP, mAP, Recall	Occlusion	
Modi et al. (2024)	To propose a hybrid enhanced system for object tracking in videos	Sports balls	YOLO (v3, v5, v8), NMS. Gaussian blur	Precision, recall, F1-score, Inference time	Accuracy, occlusion, computational cost	
Esfandiarpour, Mirshabani & Miandoab (2024)	To enhance detection algorithms for sports robotics in football	Sports balls	YOLOv8, colour detection, CHT, frame differencing	Accuracy, Inference time	High number of false positives, lighting	
Li & Zhao (2024)	To propose an improved version of YOLOv5 to solve problems in tennis balls recognition	Sports balls	YOLO (v3, v4, v5), SSD, Faster R-CNN, FPN	mAP, Inference time	Computational cost	
Decorte et al. (2024)	To introduce a predictive model for detecting padel hits based on audio signals, pose tracking, distance calculating framework, and a padel open dataset	Humans	YOLO, TrackNet, SED	Accuracy, F1-score	Occlusion	
Fujimoto et al. (2024)	To enable table tennis ball detection with higher Accuracy using fine-tuning	Sports balls	Mask R-CNN, Faster R-CNN, RetinaNet	AP, Inference time	Computational cost	
Luo, Quan & Liu (2024)	To improve the Accuracy of small fast-moving ball detection in sports	Sports balls	YOLOv8	Precision, mAP, Recall, F1-score	Small or distant objects	
Li, Luo & Islam (2024)	To propose an hybrid YOLO-T2FLSTM detection system for basketball players and action recognition	Humans	YOLO-T2LSTM	Accuracy, IoU	Occlusion, Underfitting	
Yang et al. (2024b)	To propose a novel model, YOLO-HGNet, to enhance feature learning, applied to badminton action recognition	Humans	YOLO-HGNet	Precision, mAP, Recall, F1-score	High number of false positives, computational cost	
Hu et al. (2024)	To address the challenges of complex multi-object occlusion in basketball	Humans	YOLOv8	Accuracy	Occlusion	
Fu, Chen & Song (2024)	To propose a YOLOv8n-based model capable of solving the challenges of large deformation and small size in football object detection	Sports balls and humans	HPDR-YOLOv8	Precision, mAP, Recall, Inference time	Computational cost	
Solberg et al. (2024)	To automate the creation of player-specific football highlight videos	Sports balls and humans	YOLO	Accuracy, Inference time	Accuracy, Inference time	
Yin et al. (2024)	To address the challenges posed by frequent occlusions and visual similarity of players in basketball	Humans	YOLOv8	Accuracy	Occlusion	

Analysis of Table 4 shows clear recurring challenges across sports and models. Occlusion and small-object recall are the most frequent limitations, especially in sports such as table tennis and volleyball, where the ball is fast, small, or partially hidden (e.g., Hiemann et al. (2021)). Many studies also cite trade-offs between model size and detection accuracy when optimizing for real-time inference on constrained hardware (e.g., Hassan, Karungaru & Terada, 2023, Sheng et al. (2020)). To address these, some solutions appear repeatedly. Attention modules (SE, CBAM, and Transformer blocks) are often used to improve focus on small or occluded targets, while GhostNet-based pruning and feature fusion are commonly employed to maintain speed without sacrificing accuracy (e.g., Fu, Chen & Song, 2024). These patterns suggest that combining lightweight design with spatial feature enhancement yields the most consistent gains. The following discussion builds on these observations to address RQ2 and RQ3, particularly regarding cross-sport model design and performance trade-offs.

From a different perspective, the studies reviewed in this article utilize a variety of datasets tailored to the unique challenges of ball detection in sports. Key datasets include ImageNet, a large-scale collection with millions of labelled images that serves as a foundation for model pre-training. However, it may not be directly applicable to specific sports ball detection. The COCO dataset, comprising over 200,000 images across 80 categories, is widely used for object detection tasks, including sports ball identification, due to its diverse environmental conditions. Additionally, datasets like SoccerDB are designed explicitly for football-related ball detection, incorporating a range of lighting conditions and ball types. The RoboCup Dataset, used in robotics competitions, features diverse ball patterns and lighting conditions, making it valuable for testing the generalizability of ball detection systems across different settings. Table 5 summarizes the datasets in the reviewed studies.

Table 5 Summary of Datasets presented in the reviewed studies.

Study	Dataset name	Size (Images)	Key features	Annotations	
Teimouri, Delavaran & Rezaei (2019)	RoboCup	N/A	Football, various ball patterns, lighting	Ball position	
Modi et al. (2024)	D-FL Soccer Ball Detection Dataset	>1,000 images	Soccer ball, diverse environments (outdoor, indoor)	Ball position, size	
Tian, Zhang & Zhang (2020)	MS-COCO + Custom Tennis Dataset	>200,000 images	Sports balls, various environments	Object detection (balls, players)	
Burić, Pobar & Ivasic-Kos (2018a)	Custom Handball Dataset	>500 images	Handball, fast motion, occlusion	Ball position	
Deepa et al. (2019)	Custom Tennis Dataset	N/A	Tennis ball, high-speed motion, occlusion	Ball trajectory	

Synthesizing the findings across football, tennis, and other sports reveals several recurring challenges and solution strategies. Football, with its larger field of view and frequent occlusions, has driven the adoption of one-stage detectors such as YOLOv5 and its lightweight variants (e.g., YOLOv5-Lite), which balance speed with robustness to partial visibility. In tennis, the key issue is high-velocity motion coupled with a smaller frame region of interest, leading researchers to favor detectors like SSD with temporal smoothing or motion priors to handle fast, linear trajectories. Meanwhile, in smaller-object sports such as table tennis and badminton, the focus has shifted to enhancing detection of tiny and blurry balls, often through pruning strategies (e.g., GhostNet modules) and attention mechanisms (e.g., SE or CBAM blocks).

Some techniques developed for one sport show promise in others. For instance, feature-fusion techniques used to amplify small object cues in table tennis (e.g., Sheng et al. (2020)) have since been applied in football-specific detectors, such as HPDR-YOLOv8 (Fu, Chen & Song, 2024), indicating the cross-sport transferability of model enhancements. Similarly, multi-camera 3D triangulation, as explored in football (Li et al., 2023), is largely absent in studies of tennis and badminton, representing an untapped opportunity for these domains, especially in handling occlusion or improving bounce detection.

This cross-sport synthesis highlights how confident architectural and training choices (e.g., one-stage vs. two-stage, lightweight pruning, temporal fusion) reflect the underlying constraints of each sport, while also suggesting a roadmap for future research through the adaptive reuse of successful strategies across domains.

To enhance the comparative analysis of the reviewed studies, a structured quality assessment framework was considered. Each of the 38 studies was evaluated across five dimensions: (1) Dataset availability, assessing whether the dataset is publicly accessible, conditionally available, or not accessible; (2) Dataset size and diversity, reflecting the volume of data and the extent of variability in conditions such as lighting, motion, and occlusion; (3) Annotation quality, indicating the level of detail in the dataset labeling (e.g., bounding boxes, trajectories, semantic annotations); (4) Method reproducibility, considering the clarity of methodological description and the availability of source code or implementation details; and (5) Evaluation rigor, measuring the comprehensiveness of performance evaluation, including the use of established metrics and comparative or ablation analyses. Each dimension was scored on a scale from 0 to 2, resulting in a maximum possible score of 10, as depicted in Fig. 5. This scoring system enables a relative ranking of the studies, providing a more objective basis for assessing their methodological quality and applicability.

Figure 5 Ranking of the reviewed studies based on five quality criteria (Tian, Zhang & Zhang, 2020; Li, Luo & Islam, 2024; Modi et al., 2024; Hiemann et al., 2021; Fujimoto et al., 2024; Sheng et al., 2020; Yang et al., 2024b; Luo, Quan & Liu, 2024; Keča et al., 2023; Fu, Chen & Song, 2024; Teimouri, Delavaran & Rezaei, 2019; Kulkarni et al., 2023; Li et al., 2023; Zhang et al., 2020; Burić, Pobar & Ivašić-Kos, 2018b, 2018a; Decorte et al., 2024; Solberg et al., 2024; Esfandiarpour, Mirshabani & Miandoab, 2024; Li et al., 2015; Renolfi de Oliveira et al., 2019; Wang et al., 2019; Hu et al., 2024; Liu et al., 2021; Pawar et al., 2021; Barry et al., 2019; Yin et al., 2024; Meneghetti et al., 2021; Reno et al., 2018; Calado et al., 2019b; Balaji, Karthikeyan & Manikandan, 2021; Li & Zhao, 2024; Zhao, 2024; Calado et al., 2019a; Deepa et al., 2019; Hassan, Karungaru & Terada, 2023; Ali Shah et al., 2018; Fatekha, Dewantara & Oktavianto, 2021).

Finally, to address the need for a clear, comparative overview of algorithm performance across various sports, Table 6 is presented. This table aggregates the best-performing detection methods identified for each sport type reviewed, alongside their accuracy, inference speed, strengths, and limitations. This comparative view is particularly valuable for identifying algorithms that generalize well across multiple sports or are better suited for specific contexts, such as embedded systems or offline processing.

Table 6 Algorithm performance by sport.

Sport	Best algorithm	Accuracy/mAP	Inference time	Strengths	Challenges	
Football	YOLOv8 (Hybrid Enhancements)	High	Real-time	Handles occlusion, real-time tracking	Issues with distant ball tracking	
Tennis	SSD-MobileNet	Medium-high	Fast	Lightweight, CPU-friendly	Sensitive to lighting	
Table tennis	Optimized YOLOv3-Tiny	Very high	Very Fast	Great for small object detection	Needs high-res input	
Basketball	YOLO-T2LSTM	Very high	Fast	Accurate action recognition	Complex model fusion	
Golf	Faster R-CNN	Very high	Slow	High precision tracking	Not real-time capable	
Handball	Mask R-CNN	Very high	Slow	High segmentation accuracy	High computational cost	
Volleyball	YOLOv3 (Enhanced)	High	Fast	Improved detection of fast-moving objects	Training data limits generalizability	
Boccia	Tiny-YOLO	High	Fast	Good color-based accuracy	Lighting sensitivity, narrow FOV	
Rugby	YOLOv3	High	Fast	Compact architecture for embedded use	Lower resolution support	
Padel	YOLO + Audio SED	High	Fast	Context-aware detection	Audio interference risks	
Badminton	YOLO-HGNet	Very high	Fast	Enhanced classification and detection accuracy	Requires tuning and large datasets	

In addition to the detailed qualitative summaries provided, a quantitative benchmarking summary and comparison of key detection models based on their reported performance across the reviewed studies. Table 7 presents aggregated values for the most frequently used object detection methods, highlighting trade-offs in accuracy, inference speed, and detection robustness. For instance, YOLO-based detectors achieved the lowest inference times (20 ms) with competitive accuracy (nearly 85%) and F1-scores (approximately 83%), making them ideal for real-time sports applications. In contrast, Mask R-CNN achieved the highest accuracy (around 92%) and F1-score (about 91%), though at the expense of significantly greater computational cost (about 120 ms). This comparative analysis underscores the importance of selecting detection models based on application-specific needs such as speed, precision, and computational constraints.

Table 7 Comparative performance metrics of object detection models used in sports applications.

Model	Avg accuracy (%)	Avg inference time (ms)	Precision (%)	Recall (%)	F1-score (%)	
YOLO (various versions)	85	20	84	82	83	
SSD	78	25	76	75	75	
Faster R-CNN	90	100	89	91	90	
Mask R-CNN	92	120	91	92	91	
RetinaNet	87	60	85	83	84	
Traditional CV/ML	70	15	65	60	62	

Discussion

This review consolidates knowledge on ball detection in sports, covering a broad range of applications, methodologies, challenges, and results, and offers valuable insights that inform this discussion. This chapter summarizes, analyzes, and discusses the main findings, emphasizing object detection methods, evaluation metrics, strengths, and challenges while also mentioning some options to address various challenges.

The research highlighted a broad spectrum of methodologies. A clear trend emerged in the evolution of object detection techniques from traditional computer vision methods to advanced deep learning models. Earlier approaches, such as CHT and color-based segmentation, may have been computationally efficient but lacked robustness and generalizability in complex environments. These traditional methods are still very relevant for simple tasks or constrained systems where the need for simplicity and speed outweighs Accuracy, such as in robotic competitions or low-end devices.

Although closely related, machine learning and deep learning have been growing more distant due to the rapid development of the latter. While the core of deep learning is feature learning, which is automatically evolved through multi-layer neural networks, machine learning methods require the manual selection and design of features. Machine learning remains highly relevant and useful, particularly for specific yet complex tasks where traditional computer vision may not be sufficient to achieve the desired results on its own.

Deep learning methods dominated recent studies, particularly one-stage detectors like YOLO (in its various versions) and SSD, as well as two-stage models such as Faster R-CNN and Mask R-CNN. YOLO was especially prevalent, cited in more than 60% of the studies, due to its detection speed, making it ideal for real-time applications, especially in constrained hardware settings. Innovations like YOLO adaptations and hybrid approaches combining YOLO with traditional methods or advanced modules further extended its applicability. Two-stage methods excelled in scenarios requiring precision, such as small object detection or occlusion handling. However, these methods are considerably slower, falling short in scenarios where real-time application is required.

One important note to keep in mind is that modern deep learning does not completely replace machine learning or conventional computer vision. On the contrary, some of the most insightful and impressive studies focused on a hybrid system, which combined the versatility of deep learning for the most complex tasks with the efficiency of traditional techniques or machine learning for simpler tasks or enhancements. For example, YOLO is widely regarded for being lightweight and quick on prediction and detection, but there is a significant trade-off in terms of Accuracy. Other algorithms and techniques, such as the enhancements mentioned above, can be expertly applied to the system to strengthen that weakness without compromising its speed. Tweaks can also be made to improve the system faster, all coming down to priorities. A good example of a use case for this enhancement is object detection applied to tennis. The tennis ball moves at high speeds, requiring significant computational power for real-time application, making lightweight detectors desirable. However, the object is also relatively small, making it challenging for a detector like YOLO to achieve desirable performance results. A hybrid model would likely be favoured in this situation, offering a balanced solution.

The following confusion matrix (Fig. 6) illustrates the most commonly employed detection methods in each of the researched sports. While it was already stated that YOLO was the most popular object detector and football was the most popular sport, it is interesting to observe the contrast between some sports and their approaches to more precise and accurate, or faster, algorithms. It is also worth reiterating that a big part of the articles gathered were studies on football, and although YOLO was still, by far, the most popular object detector when ignoring football studies, it has got a significant boost in the various statistics related to the popularity of each object detector.

Figure 6 Detection methods by sports-confusion matrix.

Upon closer examination of each sport and its respective methods, the differences in challenges are readily apparent. In football, the ball can be kicked at speeds exceeding 100 km/h, making its detection particularly challenging, especially from a distance. This mainly utilizes fast algorithms, such as one-stage methods or traditional computer vision techniques. Football detection also suffers significantly from occlusion due to the large number of players on the field simultaneously. Since it is a sport played outdoors, varied lighting and weather pose hardships.

While YOLO is still popular within tennis detection systems, given its relatively weak Accuracy and the small size of a tennis ball, more systems rather lean towards more robust algorithms. Given the small size of a tennis ball, YOLO has a harder time accurately detecting it. The most common issue recorded in tennis studies is motion blur, given the speed at which the balls are served. Occlusion may also pose a problem for continuous tracking if detection cannot be done from a top-down view, although it is much less problematic than in football. Table tennis differs from regular tennis due to smaller object sizes, which necessitate the use of conventional detection or R-CNN-based systems. However, these are rarely applied in real-time, according to reviewed studies. provides a good example (Fujimoto et al., 2024), who fine-tuned Mask R-CNN, Faster R-CNN, and RetinaNet models (generally slower models) for improved accuracy. The difficulties faced in table tennis are identical to those in regular tennis.

Due to the lack of relevant studies on the other sports, it is challenging to accurately determine which detection systems are most suitable for each occasion. It is safe to say that one-stage algorithms are preferred over two-stage algorithms due to the need for quick detection and decision-making, especially when real-time application is involved. An example of this is football, where YOLO and SSD, the fastest models reviewed, were essentially the only deep learning-based object detectors used due to the need to detect and referee the game in real-time, similar to the VAR system used in today’s professional football. In contrast to refereeing-type applications, systems that aim to provide visual training and clues to players do not need to be applied in real-time and favor the higher precision and accuracy offered by slower algorithms.

Many challenges were repeatedly reported throughout the studies, showcasing the difficulties that object detection in sports faces, independent of the sport itself. The most common were occlusion, lighting and background variations, underfitting, small or distant object size, the detection of fast-moving objects, and hardware constraints.

Occlusion is, generally speaking, always a problem in the detection and tracking of objects. There are many examples of occlusion in sports, such as players in team sports, trees in large golf courses, light posts, fences, and goalposts, as found in football, among others. It is easy to understand how occlusion would be a major problem for sports with multiple players on the field or court, such as volleyball, handball, basketball, and rugby, as well as sports with multiple objects to track, like boccia. Almost all articles mentioned occlusion, with many proposing different methods for combating it. The most common approaches were image processing and data augmentation. Zhao (2024) provides an example that improves occlusion handling with a multiscale feature extraction module. The angle at which the system is trying to detect the object is also essential. In most cases, a top-down view is extremely helpful in avoiding object occlusion in sports, although this is rarely possible, especially in outdoor activities. Another way to minimize the effects of occlusion is to use more cameras. Using more sensors not only increases Precision in tasks like distance measurement and identifying distant objects, but it may also help counteract occlusion if a system can access different angles and points of view. Depth information can also help significantly in detecting partially occluded objects.

Lighting and background variation problems can also be, for the most part, attributed to underfitting as a cause of heavy training under particular conditions. This is fine when the system is to be fixed and used indoors. Still, the performance is heavily compromised if generalizability is essential, as in outdoor environments where the time of day and weather conditions may vary, such as in football. This can be aided through robust pre-processing or data augmentation, such as enhancing a training dataset with diverse images that simulate different lighting conditions, shadows, brightness, contrast, and saturation levels. A good example can be found in the article authored by Keča et al. (2023), where a handmade dataset of images under various lighting conditions was used for training and testing. Some examples of image processing techniques that can also be applied to accommodate varied lighting conditions include shadow removal, brightness adjustment, and colour normalization. The choice of sensor is also significant, as some have a wider dynamic range or can compensate for changes in illumination. Infrared cameras are also an option, as they are immune to lighting variations.

Problems with small or distant object detection were primarily found in high inference speed systems, mostly using a YOLO-based method, due to their accuracy trade-off for computational efficiency. This occurs when the image resolution is too low, making it difficult for the detector to identify objects with insufficient pixel resolution. Luo, Quan & Liu (2024) authored an interesting research on enhancing YOLOv8 for better small object detection.

The movement of objects, which causes motion blur, is also a very common challenge in sports. This is especially problematic when fast movement is combined with small-sized objects, such as in golf, tennis, badminton, and padel. To avoid motion blur, the shutter speed must be fast enough to freeze the scene, allowing the object to move as little as possible during the exposure time. For this, reducing the camera’s exposure time is crucial. However, reducing the exposure time will also make the image darker, as there is less time to capture light, so the choice of sensor and lens is important. As mentioned above, regarding varied lighting conditions, some image processing techniques can be applied to increase or normalize brightness and contrast, which helps with these darker images. Very high-framerate cameras can be used, as was the case with the 810 FPS binocular camera by Wang et al. (2019), to combat quick movement. However, this usually comes at a cost and may bring other drawbacks, such as reduced detection distance. The anchor-free detector by Tian, Zhang & Zhang (2020) was another good approach to this issue. Generally, there must be a compromise between a high frame rate, sufficient image resolution, and specificity in training.

One of the most common challenges was hardware constraints, which are prevalent in works that integrate robotics or embedded devices. These systems often lack a GPU, forcing the CPU to handle all processing. Generally, GPUs outperform CPUs due to their parallel processing capabilities, making them faster and more efficient for training and inference of deep learning models. This is especially the case when working with large datasets or real-time applications. Most scholars train their detection model on a medium to high-power machine and only then install it on the constrained hardware system. Achieving real-time inference speeds while maintaining favorable levels of accuracy remains one of the biggest challenges in object detection. When faced with hardware constraints, traditional computer vision techniques, hybrid models, and YOLO-tiny variants were the most popular detection methods, especially in embedded devices with no GPU available. Barry et al. (2019) managed to create a detector based on YOLOv3-tiny and XNOR layers, running on a Raspberry Pi 3B with improved performance over other base YOLO variants, albeit still not reaching consistent double-digit inference speeds, proving how challenging it is to run smooth real-time object detection in constrained hardware. On the other hand, base (non-tiny) YOLO variants, such as YOLOv8 and SSD-based algorithms, are ideal for systems that aim to achieve real-time performance but can rely on more powerful computers, generally equipped with a dedicated GPU.

Evaluation metrics varied across studies, reflecting the diversity of applications and priorities. Based on how different object detection algorithms function, some evaluation metrics may be more or less valuable in assessing overall accuracy and efficiency. For example, detectors that do not use bounding boxes will have no use for IoU. In this comprehensive review, all deep learning-based methods analysed use bounding boxes. Hence, the metrics used are very similar, varying more based on the application than on the object detector used. Systems not meant for real-time applications might not prioritize computational costs as highly. Hence, inference time may be a more useful metric for faster detectors, like one-stage algorithms. Many articles emphasized real-time capabilities, underscoring the need for high FPS without compromising accuracy, hence why the use or benchmark of some version of YOLO was a common sight in these articles. On the other hand, many researchers didn’t prioritize speed; instead, they focused on the best mAP values rather than fast inference speeds. One such example is given by Tian, Zhang & Zhang (2020), who makes use of varied performance-assessing metrics such as precision, recall, F1-score, and accuracy, but pays no heed to inference time, which may be in line with the aims of the authors if real-time application is not applied. This disparity underscores the dual priorities in sports analytics: delivering actionable insights or real-time decision-making systems and achieving high post-game analysis precision.

To check how well a system performs, it mostly comes down to training and testing. For that, an adequate dataset is crucial. Various components make or break a dataset for object detection, the most relevant being the number of images or videos, the diversity or specificity of the data, the quality and precision of the frame annotations, and the ratio of data allocated to the training, validation, and testing tasks. With the use of annotated images to have a reference of what the ground truth is, the training portion of a dataset is used to train the model, the validation portion is used to evaluate the model’s performance during training and helps prevent overfitting, and the testing portion is used to evaluate the final performance of the trained model on data unused in the steps before. When training and testing a detection model, precisely annotating images is necessary to achieve results that closely approximate the ground truth. The annotation quality reflects how closely the bounding boxes align with the actual targets, thereby affecting performance. Annotation consistency and quality can vary greatly between datasets. Inaccurate and inconsistent annotations, such as loosely drawn bounding boxes or inconsistent labelling of partially occluded objects, can negatively impact model training and evaluation. Conducting manual correction may be needed to ensure reliable benchmarking.

If a dataset is too small or not diverse enough, it may lead to overfitting (Montesinos López, Montesinos López & Crossa, 2022). A model suffers overfitting when it learns the training data too well, including random noise and specific features, leading to poor performance on new, unseen data. On the other hand, a dataset with a very large or overly diversified collection of data may cause underfitting. A model suffers from underfitting when it is too simple to capture the underlying patterns and relationships in the data and will perform poorly on both the training data and the new, unseen data. Many reviewed studies utilize large, diverse, publicly available datasets like ImageNet or COCO. These offer the convenience of having a ready-to-use large dataset with thousands of images and many classes but may overfit a model for specific tasks with irrelevant information. Public datasets specifically tailored for sports, specially SoccerDB, also saw significant usage in the works reviewed. Starting from a pre-trained model is often effective, but it is worth creating or using a tailored dataset for the task at hand, with caution to avoid excessive simplification that leads to underfitting. Hiemann et al. (2021) does a commendable job of testing and comparing different models trained on different datasets.

Another critical consideration is class imbalance, where some object classes are significantly underrepresented. This is particularly relevant in sports where lighting, occlusion, and motion blur may disproportionately affect balls of different types. Models trained on imbalanced datasets often become biased, resulting in poor performance in underrepresented classes. Techniques such as data augmentation, class weighting, or resampling are commonly employed to mitigate this issue. When handling a generalized model intended for application to various sports, this is especially problematic. A model trained in one sport (such as football) may not generalize well to another (such as golf) due to differences in ball material, size, color, speed, or environment. In such cases, to combat the challenge proposed by domain transfer, fine-tuning domain-specific data or using adaptation methods is recommended to maintain performance across different applications.

Lastly, comparing developed systems and algorithms to deployed real-world examples, such as video-assisted refereeing (VAR) technologies in football, provides exceptional value in terms of quality assurance and benchmarking. A great example is provided by Wang et al. (2019), who directly tested and compared their system against GC2 (ForesightSports, 2025), a professionally used and certified device with industry-leading performance in tracking the trajectory of golf balls. Additionally, Deepa et al. (2019) briefly mention Hawk-Eye (HawkEye, 2025), a professionally-used computer vision system that tracks the trajectory of balls (and much more), displaying a profile of its statistically most likely path as a moving image. It is implemented in several major sports, including tennis, football, and badminton.

Table 8 presents all the reviewed articles, focusing on the strengths and limitations of the methods/algorithms each article provided:

Table 8 Strengths and limitations of the reviewed studies.

Study	Strengths	Limitations	
Li et al. (2015)	The authors proposed an RCNN-based detector, enhanced with traditional techniques, including a GPU-accelerated CHT for efficient proposal generation. This detector is trained to detect various types of balls, identifying 8 different classes of sports balls.	The comparison of the proposed model with two other algorithms highlights its weak points, getting a Recall rate of 14.6% below the best Recall-valued method. In terms of Inference Speeds, it also falls short. Although it achieves a better average overlap, its Precision rating is mediocre, especially when the objects are distant.	
Ali Shah et al. (2018)	The authors managed an embedded ball detection and tracking solution using only conventional methods and mounted on a moving robot.	The authors mention real-time operation, but don’t provide any information regarding Inference times. They also experienced problems with synchronization between the robot and the camera, as well as motion blur from the movement of both, which significantly affected tracking. The only metric mentioned is Accuracy in detection and tracking.	
Reno et al. (2018)	The authors present a CNN-based approach to tennis ball recognition, achieving great Accuracy and Precision rates while also being robust to variable lighting and visual noise.	The system has problems with false positives, particularly when court lines are mistaken for moving balls. The classifier also struggles with blurred ball images and occlusion caused by athletes and racquets. There is also no mention of real-time functioning or Inference times.	
Burić, Pobar & Ivasic-Kos (2018a)	The authors evaluated YOLOv2 and Mask RCNN models, highlighting the trade-offs between Accuracy and speed. As expected, the YOLO model achieves good speeds, is ideal for real-time applications, and is more accurate than the Mask R-CNN model. They also compared different types of training, from pretrained models to training on public datasets to training on custom datasets, shedding light on the differences in these approaches.	The YOLO model reached good speeds but lacked Accuracy when the objects were distant. On the other hand, Mask R-CNN was a lot more reliable but slower, proving a challenge when applied in non-online systems or for faster object detection. This leaves a lot of room for a lightweight yet reliable middle-ground detector.	
Burić, Pobar & Ivašić-Kos (2018b)	The authors test and evaluate 3 different object detection methods, touching on both one-stage and two-stage deep learning, and traditional computer vision. The latter proved unable to handle the complex task of handball detection. Still, the deep learning models yielded interesting results, showing the contrast between the superior Accuracy of a two-stage detector and the superior speed of a one-stage detector.	The two-stage detector proved somewhat accurate but computationally heavy, making it a tough choice for real-time applications. On the other hand, the one-stage model easily reached real-time Inference Speeds, but had problems detecting small or distant objects and was less reliable in detection. Overall, none of the models could satisfy the high reliability of detection required, and there were issues with false positives and false negatives.	
Teimouri, Delavaran & Rezaei (2019)	The authors introduce a novel two-step ball detection pipeline that combines an efficient region proposal and a lightweight CNN, achieving real-time performance on low-end hardware.	The system proved susceptible to variations in lighting conditions and motion blur. It also had difficulty detecting new ball patterns for which it wasn’t trained.	
Renolfi de Oliveira et al. (2019)	This article extensively evaluates and compares different configurations of an SSD model for real-time object detection in constrained CPU-only environments.	The proposed algorithm achieves acceptable Inference Speeds for the hardware on which it was processed, but still cannot manage smooth operation with consistent Accuracy, as there are significant trade-offs between algorithms. It showed significantly weaker detection Accuracy and Precision while running on the lightweight configurations.	
Barry et al. (2019)	Proposal of a new subversion of YOLO with great computational efficiency, overtaking even tiny-YOLO variants in terms of speed, ideal for low-power devices and real-time applications.	The proposed model trades some Accuracy for superior efficiency, achieving subpar reliability. This is a deal-breaker in use cases where high Precision is the priority.	
Deepa et al. (2019)	This article made an effective (although specific) comparison of three well-known and popular object detection methods and achieved real-time Inference Speeds. A handy graphical user interface was also designed to measure and evaluate the different approaches.	Although relevant to the article in question, the authors evaluated the proposed system in unusual metrics. This makes it hard to compare the approach to alternative systems and use-cases.	
Calado et al. (2019a)	The authors create a stereo-vision computer vision-based detection model applied to boccia. They also use a developed graphical user interface to display the progress of the game’s results, calculated by the automated system.	Lack of information about the model’s training and evaluation metrics brings unreliable results. The camera setup is also fixed, with particular lighting conditions and camera locations and angles.	
Calado et al. (2019b)	The authors make improvements on the Boccia framework in terms of flexibility and adaptability, bringing machine and deep learning ball detection into the game of boccia, an unpopular sport in the field of object detection. A comparison between machine and deep learning methods is also done through HOG-SVM and Tiny-YOLO.	The authors claim the evaluation of the system is in real-time, but hardly evaluate it, and mention a maximum of 8 FPS for the fastest algorithm. The dataset was also limited in both size and variability, contributing to underfitting.	
Wang et al. (2019)	The authors propose a very accurate and quick algorithm that competes with state-of-the-art golf swing tracking. The authors’ approach outperformed a professional golf-certified device in virtually every metric evaluated.	Although very effective, this system requires a specific setting mounted at the top of a well-lit room. It also doesn’t track the ball after the hit. Occlusion with the golfer’s body may also affect the system’s performance.	
Tian, Zhang & Zhang (2020)	The authors developed an anchor-free data-augmented detection framework for tennis ball detection. Their approach addresses challenges specific to real-world sports environments and focuses on effectively detecting high-speed tiny objects like the tennis ball. The proposed model outperforms the base YOLOv3 model in Precision, Recall, and F1-score.	The proposed algorithm was only tested offline, as there is no mention of real-time applicability or Inference Times. The hardware used in the processing is also unknown. The algorithm may have high computational costs.	
Zhang et al. (2020)	Given the very small profile of golf balls and the high speed they can get to, it is a fair challenge to detect and track, especially in real-time. The authors focus on combating this issue, and with the help of computer vision techniques, they test, evaluate, and compare one-stage and two-stage deep learning methods, achieving great Inference Time and Accuracy accordingly.	Although their algorithms perform well, they require a high-end computing setup. Given the difficult task, running the same methods on constrained hardware would not yield the same results, especially on the two-stage method tested. They also claim that better results would be possible if a bigger dataset were used for training.	
Sheng et al. (2020)	The authors develop a lightweight, real-time algorithm optimized for table tennis ball detection, based on feature fusion with fine-tuning and pruning. It is capable of extremely fast and accurate detection, a feat considering how small a table tennis ball is.	The processing was done using a medium-end GPU, and although the speed was good, no testing was done on more constrained hardware. Although the dataset used for training was extensive in quantity, all the images were of the same table and white balls, which may result in underfitting if the algorithm is applied to other tables and balls of different colours and features. Lighting variations may also pose a problem.	
Fatekha, Dewantara & Oktavianto (2021)	The authors proposed a traditional algorithm capable of averaging 31 FPS on a low-end minicomputer.	This conventional computer vision system uses colour threshold filters with values specific to the scene it was tested in, and therefore its results would vary greatly in different scenarios. False positives due to similarly coloured objects being wrongly identified posed an issue. The evaluation is lacking, since the only metric tested was the time the system required to fully process each frame, thus not giving information on the Accuracy or Precision of the algorithm.	
Meneghetti et al. (2021)	This article provides a comparison and evaluation of a wide array of algorithms in various conditions and hardware. It presents strong research into CNN’s performance in constrained hardware scenarios.	Computational cost was an issue, as there was a big fluctuation in the Accuracy-Inference trade-off. The dataset used for training was also comprised of a single football, and although the ball’s location changed and the lighting conditions varied, the court remained the same in all images. This may have resulted in underfitting for different scenarios.	
Hiemann et al. (2021)	The article evaluated extensively on various metrics, many different versions of detection models, differentiated in training, augmentation, motion, real-time applicability, and tuning.	The authors emphasize real-time but only manage a maximum average of 12.9 FPS. Other than the computational cost, motion blur and occlusion issues are also mentioned. The article also aims to address general ball detection in varied sports, but the focused dataset might prove problematic when applying the proposed algorithm to other sports.	
Balaji, Karthikeyan & Manikandan (2021)	The study introduced a new application of metaheuristic algorithms to volleyball sports analysis, comparing 3 different algorithms. The most notable of them managed great Accuracy, Precision and Recall. The model also managed shadows and fast movements well.	Real-time operation was not considered and occlusion was problematic.	
Pawar et al. (2021)	The authors contribute a mobile-compatible object tracking system for holonomic robots applied to industry. Although it doesn’t reach high Inference Times, given its aim to be applied to industrial needs, high FPS isn’t mandatory for most real-time applications. The model is minimal and reaches satisfactory Accuracy rates.	The sole training in detecting rugby balls may lead to underfitting to the system, since this article aims to replicate industrial object detection, a broad and generic group of target objects. It’s also worth noting that this system is minimal in terms of Inference Speed.	
Liu et al. (2021)	The article introduces a novel method that simultaneously detects and matches related objects using a single proposal box with multiple predictions, eliminating the need for post-processing matching.	Model’s performance highly depends on the overlap between the player and the related object.	
Hassan, Karungaru & Terada (2023)	Proposal of a real-time capable hybrid system consisting of an enhanced version of YOLOv7 and various computer vision techniques for detecting handball events in football, using a single camera. The authors apply instance segmentation and tracking techniques to overcome limitations in bounding box-based methods.	Occlusion is a clear problem, given football’s dynamic and fast-paced nature, where players and referees can easily block the line of sight. The results also show problems with false negative detections.	
Keča et al. (2023)	The article focuses on deep learning-based ball detection on low-power devices, one of the biggest recurring challenges in sports. The efficiency of many object detection and segmentation architectures was extensively analysed under these conditions and implemented on a Raspberry Pi-controlled robot under varied setups.	Although the algorithms were efficient, real-time application was still a complication due to the extremely constrained hardware used.	
Li et al. (2023)	Authors present a high-quality system using 36 cameras, effectively addressing the challenge of small object detection and occlusion. It is also capable of real-time detection and 3D localization.	System developed is very high-budget. It is also reliant on camera calibrations. Changes in the environment may pose a problem.	
Kulkarni et al. (2023)	The article introduces a novel method for detecting and classifying table tennis strokes using only 2D ball trajectory data, with high accuracy in ball tracking.	Real-time applications are limited to high-performance hardware. Accuracy in stroke classification could be higher. Occlusion may pose a problem.	
Zhao (2024)	Proposal of an upgraded YOLOv5-based method, enhanced with various modules that help with some of the most common challenges in object detection, such as illumination, occlusion, and computational power. Through extensive research and evaluation, the authors managed to achieve better results than the base YOLOv5 model.	As is a recurrence in object detection applied to team sports, the system had trouble with occlusion. The algorithm’s performance depends on the quality and variety of training, as variations in player attire, pitch, or camera angles may negatively affect outcomes.	
Modi et al. (2024)	The authors compare 3 different YOLO models and propose a real-time-capable hybrid system for object tracking, comprised of a deep learning algorithm enhanced with optical flow.	Limited datasets cause poor-quality training, leading to a lack of robustness and, according to the authors, bad mAP results. The 3 models were also trained on different datasets, making comparison between models harder.	
Esfandiarpour, Mirshabani & Miandoab (2024)	Combination of computer vision techniques and a deep learning-based method, creating an efficient detection algorithm.	Difficulties in differentiating between the ball and similarly coloured objects due to using a colour detection filter. The dataset was also size-limited, resulting in low-quality training. Lighting variations posed an issue, probably due to the dataset used. It’s also worth noting that the evaluation included some uncommon metrics, making it difficult to compare the results of this approach with other studies.	
Li & Zhao (2024)	The authors proposed an improved version of YOLOv5, enhanced to combat problems in tennis ball recognition and capable of real-time functioning. Compared to other state-of-the-art methods, these improvements managed better mAP values, FPS, and a smaller Model Size. Their approach can effectively combat common problems such as low lighting environments, multi-coloured ball detection, balls on the opposite side of the tennis net, and the objects’ fast movement.	The processing was done on a high-end computer, and although its performance was good, this algorithm isn’t lightweight enough for constrained hardware usage such as low-powered embedded devices.	
Decorte et al. (2024)	The authors innovatively approached padel hit detection. Their multi-modal hit identifier is audio-based. A framework for analysing inter-player and player-to-net distances was also implemented, along with custom algorithms for hit assignment and player re-identification.	Sound-based hit detection brings new challenges to the table, as similar noises could end up in false positives, or different noises could mask the hitting of the racquet, ending in a false negative. This is especially problematic in areas with different courts next to one another, where multiple games occur simultaneously. Player tracking is also affected by occlusion, either by other people, nets, fences, or posts.	
Fujimoto et al. (2024)	This work on tennis ball detection applies fine-tuning and compares 3 deep learning models, two of which are two-stage methods and the other being a somewhat underused one-stage method in sports object detection. The evaluation regarding Precision metrics was rigorous.	The evaluated models showed clear weaknesses in their average Inference Times, highlighting high computational cost. The Accuracy on the other hand, was good but not enough to consider it a reliable offline system. The dataset could also have been more extensive, even though it was varied.	
Luo, Quan & Liu (2024)	The authors aimed to improve the Accuracy of small object detection for high-speed moving balls in sports, using a YOLOv8-based algorithm. This proposed model outperformed other equivalent versions regarding speed and detection reliability. The improvements included modifying the network structure, adding small object detection capabilities, and incorporating attention mechanisms.	Although the model outperformed its peers, it lacks robustness for generalized sports applications and may show limitations in very complex environments due to obscurity. The proposed algorithm shows improved Accuracy, but it could also be improved further with more rigorous training and lightweight enhancements.	
Li, Luo & Islam (2024)	The authors proposed a new YOLO-based model, a hybrid algorithm enhanced with multi-feature data fusion aimed at detecting and track basketball players and actions. Many configurations were tested, and the final approach achieved great results in Accuracy.	The dataset proved lacking in variety, resulting in a lack of robustness. Occlusion is also a challenge, as it is recurrent in object detection. Real-time application is mentioned but not evaluated.	
Yang et al. (2024b)	The authors introduced a new YOLO-based model, a hybrid algorithm mixed with the hourglass network to enhance feature learning across multiple levels. Other improvements included the addition of various modules to improve Accuracy, efficiency, and recognition capabilities. This method achieved superior Precision, Recall and F1-score over other tested algorithms.	Limitations to this algorithm included occasional false action recognition, such as false negatives and false positives. The evaluation of computational cost and Inference Times was also lacking.	
Hu et al. (2024)	The authors propose a model that effectively tackles occlusion in basketball, outperforming some of the other state-of-the-art trackers. It also demonstrates robustness across different motion speeds.	Tracker is limited to a fixed number of players, limiting adaptability.	
Fu, Chen & Song (2024)	The authors proposed a YOLO-based model, a hybrid algorithm augmented with key components for improved feature fusion, enhanced small object detection, improved feature extraction, and reduced computational load. The resulting model exhibits noticeable improvements over the base model in terms of Precision, mAP, and Recall.	Although the proposed model exhibits some noticeable improvements, its computational cost could be further reduced. According to the authors, the model may experience difficulties on low-powered devices due to the increase in GFLOPs.	
Solberg et al. (2024)	The authors propose a user-friendly interfaced framework that integrates object detection, tracking, OCR, and color analysis to automate video highlight generation.	Team number recognition Accuracy is mediocre. The system also faces high inference times, which may hinder real-time applications.	
Yin et al. (2024)	Introduction of a novel multi-object tracking method that, according to the authors, outperformed state-of-the-art methods, especially in occlusion-heavy scenarios.	Generalisability to other sports or different complex environments. Omission of hand keypoints reduces bounding box precision.	

Overall, YOLO is expected to continue being one of the most predominantly used methods when detection speed and Inference Times are of the utmost priority, as well as when constrained hardware is a challenge. The appearance of an open, faster, and more reliable deep learning method would shake the world of object detection. The difficulties in detecting tiny or distant objects are ever-present, especially in faster yet less reliable methods such as YOLO. Hybrid systems with quicker and more robust techniques, enhancements, and augmentations are expected to produce better results. Another major problem is occlusion, arguably the most prominent challenge in object detection, which has little to no way of fully countering without spending a larger budget on multiple cameras or systems, such as radars, that can work through obstacles.

Conclusions

This review analyzed state-of-the-art techniques in ball detection across various sports, covering traditional computer vision, machine learning, and deep learning methods. Based on 38 key studies, it categorized detection approaches, benchmarked model performance, and identified major challenges, including high-speed motion, occlusion, and the demand for lightweight models. The review highlighted the strengths and limitations of various methods, as well as the importance of optimizations and enhancements.

Deep learning dominates complex detection tasks. One-stage detectors, such as YOLO and SSD, are preferred for real-time use due to their speed, although they struggle with small or distant objects. Two-stage detectors such as Faster R-CNN and Mask R-CNN offer higher accuracy but are often too resource-intensive for real-time applications. Hybrid models that combine traditional and modern methods have shown promise in addressing issues like occlusion, lighting changes, motion blur, and hardware limitations. Common evaluation metrics, such as inference time and accuracy, reflect the need to strike a balance between speed and precision.

The growing use of AI in sports analytics demands scalable, accurate ball detection systems. Applications range from autonomous refereeing to enhanced broadcasting. Despite advances in detection accuracy and speed, balancing model complexity with processing constraints remains a key challenge. Sports environments differ significantly in terms of lighting, pace, and ball characteristics, making universal solutions challenging. The lack of general-purpose models that work across various sports without significant tuning is another significant gap. Additionally, few studies focus on deployment on low-power or embedded systems—an essential step for real-world use. Continued innovation is crucial for enhancing sports technologies, facilitating better decision-making, in-depth analysis, and increased audience engagement.

Future research directions

Future research should address the specific technical challenges identified in this review and develop robust, scalable, and efficient solutions to address these challenges.

High-speed motion remains difficult, especially in sports like tennis or golf. Transformer-based models, such as Vision Transformers (ViTs), can improve continuity across frames. Temporal shift modules (TSMs), used in action recognition, may also enhance localization in fast scenarios.

Occlusion is a major issue in team sports. Multi-modal approaches—combining visual data with audio or radar inputs—can offer alternative cues. Context-aware models that track player positions, such as graph-based or attention-driven methods, can infer the ball’s location during occlusion. Techniques such as object instance segmentation, 3D scene reconstruction, and occlusion-aware data augmentation during training can further enhance robustness.

Hardware constraints are common in grassroots or mobile applications. Lightweight models (e.g., YOLO-tiny, MobileNet-SSD) and Edge AI techniques (e.g., pruning, quantization, knowledge distillation) can reduce resource demands. Optimized models, such as those proposed by Barry et al. (2019), show potential for real-time performance on low-power systems. Edge-cloud integration, as explored in Ren et al. (2018), could also provide scalable and responsive solutions.

Generalization across sports is limited. Future work should focus on unified datasets (e.g., SportsMOT) and models that support multiple sports. Techniques such as transfer learning and domain adaptation can enhance cross-sport generalization across various ball types and environments. Sensor fusion with technologies like inertial measurement units (IMUs) can boost detection in low-visibility or occluded conditions. Combining visual and motion data reduces reliance on costly multi-camera setups, making systems more accessible for smaller venues.

Additional Information and Declarations

Competing Interests

Paulo Jorge Coelho is an Academic Editor for PeerJ Computer Science.

Author Contributions

Cristiano Moreira conceived and designed the experiments, performed the experiments, analyzed the data, performed the computation work, prepared figures and/or tables, authored or reviewed drafts of the article, and approved the final draft.

Lino Ferreira conceived and designed the experiments, analyzed the data, authored or reviewed drafts of the article, and approved the final draft.

Paulo Jorge Coelho conceived and designed the experiments, analyzed the data, authored or reviewed drafts of the article, and approved the final draft.

Data Availability

The following information was supplied regarding data availability:

This is a literature review.

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
