# Peer review of "A comprehensive review of ball detection techniques in sports"

_PeerJ Computer Science, doi:10.7717/peerj-cs.3079_

## Round 0.1 · original submission · Major Revisions

Dear Authors,

Thank you for the submission. The reviewers’ comments are now available. It is not suggested that your article be published in its current format. We do, however, advise you to revise the paper in light of the reviewers’ comments and concerns before resubmitting it.

Best wishes,

Reviewer 1 ·

Basic reporting

While the manuscript adheres to fundamental academic writing standards, it has several significant shortcomings. Most notably, the originality of this review is unclear. The authors fail to articulate how this paper differs from previous review studies and what novel insights it provides to the field.

Moreover, although the literature review appears extensive, it lacks sufficient coverage of recent studies, particularly from 2023 and 2024. Given the rapid advancements in deep learning and computer vision, failing to incorporate the latest research significantly weakens the paper’s contribution. Some references appear dated, with several key sources published before 2019. Including more recent studies (especially from 2023–2024) could strengthen the review and ensure that the paper reflects the current state of the field.

The structure of the paper is also problematic. The transitions between sections are abrupt and disorganized, particularly between the methodology and results sections. The methodology outlines selection criteria, yet it does not convincingly explain how these criteria were applied to derive meaningful conclusions.

While the figures and tables are formatted appropriately, their interpretability is questionable. Many tables present an overwhelming amount of data without sufficient explanation, making it difficult for the reader to extract key insights. The authors should include stronger interpretative commentary alongside these visual elements.

In summary, while the paper is written in professional English, it suffers from structural deficiencies, a lack of recent literature, and an unclear statement of originality.

Experimental design

The study design is weak and lacks the necessary rigor expected in a systematic review. Although the authors claim to use a systematic selection methodology, their approach is neither clearly defined nor justified.

Several critical flaws in the study design reduce the reliability of the review:

Lack of Justification for Source Selection: The authors limit their database searches to IEEE Xplore, Springer, and Scopus, ignoring other major sources like Google Scholar or ArXiv. They do not provide any explanation for this restriction, which raises concerns about the completeness of the literature review.

Inadequate Selection Criteria: While the PRISMA methodology is mentioned, the inclusion and exclusion criteria are poorly defined. It is unclear how the quality of included studies was assessed. Simply filtering by keywords and publication date is insufficient for ensuring a high-quality review.

Absence of Bias Assessment: There is no mention of any bias assessment method, which is critical in a review study. Many included papers focus disproportionately on football and tennis, while other sports receive minimal attention. The authors should explain whether this imbalance is due to research availability or their selection criteria.

Lack of Replicability: The methodology is not described in enough detail for replication. The authors do not specify how they handled duplicate studies, conflicting findings, or inconsistent methodologies across the reviewed papers.

A well-structured study design should ensure that the literature selection is unbiased, comprehensive, and replicable. Unfortunately, the current approach lacks transparency and rigor, making the findings questionable.

Validity of the findings

The conclusions presented in the paper are not critically evaluated and lack depth. The authors merely summarize existing literature without offering a thorough discussion of the strengths and weaknesses of different approaches. Key issues include:

The discussion on deep learning models lacks a critical evaluation of their computational efficiency versus accuracy. Many models achieve high accuracy, but at what cost in terms of inference speed and hardware requirements?

GPU vs. CPU performance is mentioned superficially, yet no concrete analysis is provided regarding which methods are practical for real-time applications.

The paper highlights several challenges in ball detection (e.g., occlusion, motion blur, varying ball sizes), but it fails to propose tangible solutions. Future research directions should be discussed in detail, rather than merely listing challenges without meaningful insights.

Overall, while the paper successfully compiles previous studies, it does not provide any substantive analysis or critique, making its findings weak and inconclusive.

Additional comments

This paper presents a superficial literature review with significant methodological and analytical flaws. Although it compiles a large number of studies, it lacks a clear contribution to the field and does not provide critical insights into ball detection techniques.

To make the paper publishable, the authors must make the following major revisions:

Clearly define the paper’s originality: Explain how this review differs from previous ones and what new perspectives it brings to the field.

Include more recent studies: The exclusion of 2023 and 2024 papers is a serious oversight. Many references are outdated, with key sources published before 2019. Updating these references will strengthen the relevance of the review.

Improve the methodology: The selection criteria must be rigorously justified, and study inclusion should be more balanced across different sports.

Provide in-depth analysis: Instead of merely listing existing work, the authors must critically evaluate the advantages and limitations of different approaches, particularly regarding computational efficiency, real-time feasibility, and practical applications.

Strengthen the study design: The methodology must be transparent, detailed, and replicable, with a clear justification for source selection and bias assessment.

Strengthen the conclusions: The paper should not just highlight challenges but also propose potential solutions and future research directions.

Cite this review as

Reviewer 2 ·

Basic reporting

1. The manuscript mentions the difficulty of generalizing ball detection methods across different sports but lacks an in-depth analysis of sports-specific constraints (e.g., ball visibility in baseball vs. table tennis). Please expand the discussion on sport-specific adaptations and include more examples of methods that perform well across multiple sports.

Experimental design

2. While several performance metrics (e.g., mAP, Recall, IoU) are mentioned, the paper does not compare these metrics across different detection models in a structured way. Consider including a comparative table summarizing detection models, their strengths/weaknesses, and performance metrics to offer a clearer overview for readers.

Validity of the findings

3. The manuscript highlights real-time ball detection as essential but does not thoroughly address the trade-offs between accuracy and inference speed in practical scenarios. Please elaborate on how different models (e.g., YOLO, SSD) handle real-time constraints and whether hybrid models could offer a balanced solution.

Additional comments

4. Although benchmark datasets are mentioned, the paper does not critically assess how these datasets influence model performance across various sports environments. Please provide a discussion on dataset limitations (e.g., dataset bias, insufficient diversity) and propose recommendations for more robust, sports-diverse datasets.
5. The future research directions section briefly mentions optimization and sports analytics but lacks concrete suggestions. Please expand this section to propose specific innovations (e.g., transformer-based models, lightweight edge computing adaptations) and discuss how emerging technologies could improve detection in complex sports environments.

Cite this review as

·

Basic reporting

In the study titled "A comprehensive review of ball detection techniques in sports," traditional computer vision techniques and modern deep learning methods applied to ball detection in sports are analyzed and compared. To enhance the clarity and effectiveness of the study, the following revisions are recommended:
- Data analysis is insufficient, and the use of more comprehensive statistical analysis methods is advised.
- The language used in various sections needs to be clearer. Phrases in lines 77, 121, and 128, among others, make comprehension difficult.
- The proposed model has been tested only offline, with no information provided regarding its real-time applicability.
- High computational costs and low FPS rates on limited hardware may hinder the practical use of the model. Optimizations can be made to improve computational efficiency and ensure better performance on low-power devices.
- The proposed model occasionally experiences incorrect action recognition issues. Additional validation mechanisms and improved data augmentation techniques can be employed to reduce the algorithm's error rate.
- Image processing techniques may fail under certain lighting conditions and motion blur. Measures to make image processing techniques more adaptable and to improve performance in low-light conditions should be clarified in the study.
- The metrics used in comparing the proposed model with other studies are difficult to understand. Therefore, it is recommended to utilize standard metrics in comparison procedures.

Experimental design

no comment

Validity of the findings

no comment

Cite this review as

---

## Round 0.2 · Minor Revisions

Dear Authors,

Although one reviewer accepts the revised manuscript, one reviewer suggests minor revision. We encourage you to address the concerns and criticisms of Reviewer 1 and resubmit your paper once you have updated it accordingly.

Best wishes,

Reviewer 1 ·

Basic reporting

The authors have made several positive revisions to enhance readability, structure, and clarity of figures/tables. However, several shortcomings remain:

Originality and Contribution: Although the authors state that the manuscript’s novelty is now emphasized in multiple sections, the claim remains vague and somewhat superficial. The distinction between this review and previous literature is not convincingly articulated. In particular, Table 1 still lacks a clear narrative framing that would highlight the novelty of the current review in terms of methodology, taxonomy, or scope.

Literature Currency: The inclusion of only one new study (from 2025) despite conducting an updated literature search is insufficient given the pace of advancement in deep learning and computer vision. The justification that only one relevant paper was found raises questions about the breadth of the search strategy, especially since sources such as ArXiv and more inclusive indexing engines (e.g., Google Scholar) were not fully utilized.

Response Letter Clarity: Several responses in the final "Additional Comments" section are marked as “N/A” or were not directly addressed. These points were not optional, as they summarize key recommendations (e.g., originality, future directions, recent literature), and should have been answered explicitly.

Experimental design

The authors have correctly revised their initial claim of a "systematic review" and repositioned the manuscript as a "comprehensive review." While this is appropriate, there are remaining concerns regarding methodological rigor:

Inclusion and Exclusion Criteria: Although some updates were made to the inclusion/exclusion criteria, the description of how study quality was assessed remains underdeveloped. There is no mention of a scoring system, inter-rater reliability, or exclusion due to methodological weakness in the reviewed studies.

Database Limitations: The review relies primarily on IEEE, Springer, and Scopus. The authors note that Google Scholar was included, but no clear evidence of expanded coverage (e.g., inclusion of ArXiv) is provided. This selective sourcing reduces the comprehensiveness of the review.

Bias and Replicability: The authors now mention sport-specific bias (i.e., overrepresentation of football and tennis), which is commendable. However, there is no discussion of how such bias might affect the generalizability of findings or whether any corrective sampling strategies were considered. Also, while the authors assert that the method is replicable, a PRISMA-style flowchart or reproducible search syntax is not included.

Validity of the findings

There is a clear improvement in critical discussion, particularly in the areas of hardware constraints, model performance, and evaluation metrics. Nonetheless, the following concerns remain:

Depth of Comparative Analysis: While a new section comparing object detection models has been added, it still lacks detailed benchmarking across datasets and sports types. Readers would benefit from a clearer conclusion about which models are best suited for real-time applications under hardware constraints (e.g., YOLOv5 on embedded GPUs vs. SSD on mobile devices).

Dataset Critique: The manuscript now briefly discusses dataset limitations. However, this section is underdeveloped. There is no exploration of dataset imbalance, domain transfer challenges, or differences in annotation quality, which are crucial when evaluating ball detection across sports.

No Real-World Validation: Although the authors clarify that no model is proposed, the lack of any case study or real-world example reduces the practical relevance of the paper. A review of real-world deployments (e.g., Hawk-Eye, VAR systems) could enhance applied value.

Additional comments

Several critical suggestions (from Reviewer 1's "Additional Comments" section) were not individually addressed. Particularly, points regarding originality, methodology transparency, and practical applications were either marked as “N/A” or vaguely attributed to previous answers. A proper response would require summarizing how each recommendation was satisfied.

The manuscript still lacks a clear future directions section grounded in technical possibilities. While some emerging methods (e.g., transformers, edge AI) are briefly mentioned, they are not tied to specific challenges in the field (e.g., high-speed tracking, occlusion handling, lightweight deployment).

Minor language issues remain throughout the text. A final professional proofreading is recommended to enhance overall clarity.

Cite this review as

Reviewer 2 ·

Basic reporting

The authors have adequately addressed my comments. I recommend the publication of the revised manuscript.

Experimental design

.

Validity of the findings

.

Additional comments

.

Cite this review as

---

## Round 0.3 · Minor Revisions

Dear Authors,

While one reviewer accepts the revised manuscript in its final form, another suggests minor revisions. It is recommended that the concerns and criticisms raised by Reviewer 3 are addressed, and that the paper is resubmitted once the necessary updates have been made.

Best wishes,

Reviewer 1 ·

Basic reporting

The authors have adequately addressed my comments. I recommend the publication of the revised manuscript

Experimental design

The authors have adequately addressed my comments. I recommend the publication of the revised manuscript

Validity of the findings

The authors have adequately addressed my comments. I recommend the publication of the revised manuscript

Cite this review as

·

Basic reporting

The sections of the paper are well organized, and the methodology, along with its advantages and disadvantages, is clearly presented. Additionally, the literature review and database information are provided in detail. In addition to these strengths, the following evaluations would further contribute to the quality of the study:

• Although relevant information is presented within the text, it should be supported with more visual summaries through graphs and tables. Furthermore, some connections between the studies in the literature review are weak and should be clarified.
• The paper repeatedly refers to the same points, resulting in unnecessary repetition.
• The conclusion section should be revised to provide a more concise and straightforward summary of the study.

Experimental design

The study clearly presents the sections of introduction, related work, methodology, findings, discussion, and conclusion. However, in the related work section, there is a lack of clarity regarding the datasets used in the reviewed studies, their sizes, and the specific features contained within these datasets. For future studies, tools such as PRISMA can be utilized for quality assessment and scoring methods. This would enable the establishment of a ranking criterion for the reviewed studies.

Validity of the findings

Although performance metrics such as Precision, Recall, and mAP are used in the analysis of the findings, the overall evaluation is largely qualitative in nature. Since the hyperparameters of the algorithms used in the study are not specified, it is unclear under which conditions each algorithm performs better. Therefore, visually supported tables that allow for performance comparisons between algorithms are important. This would also help identify the most suitable algorithm for each of the more than 12 different sports types examined

Additional comments

The conducted study is an important contribution to the fields of sports analytics and image processing. However, it lacks quantitative comparisons and certain statistical analyses. These shortcomings are critical, especially for real-world applications

Cite this review as

---

## Round 0.4 · accepted · Accept

Dear Authors,

Thank you for addressing the reviewer's comments. Following a rigorous review process, your paper has been deemed suitable for publication.

·

Basic reporting

The authors have made the necessary improvements to address the reviewers' concerns. I find the manuscript worthy of publication.

Experimental design

The authors have made the necessary improvements to address the reviewers' concerns. I find the manuscript worthy of publication.

Validity of the findings

The authors have made the necessary improvements to address the reviewers' concerns. I find the manuscript worthy of publication.

Additional comments

The authors have made the necessary improvements to address the reviewers' concerns. I find the manuscript worthy of publication.

Cite this review as